# From Basis to Basis: Gaussian Particle Representation for Interpretable PDE Operators

**Zhihao Li** [1]  **Yu Feng** [1]  **Zhilu Lai** [1 2]  **Wei Wang*** [1 2]

## Abstract

Learning fluid PDE dynamics has increasingly benefited from neural operators and Transformer-based models, but their latent states often remain opaque, and sample-wise attention can be costly at high resolutions. We propose the *Gaussian Particle Operator* (GPO), a basis-to-basis neural operator that represents fields with a learned *Gaussian basis*. Each atom carries explicit geometric parameters, including centers, anisotropic scales, and weights, yielding a compact, mesh-agnostic, and directly visualizable intermediate representation. GPO operates in modal space: learned Gaussian modal windows perform Petrov–Galerkin measurements, and a PG Gaussian Attention module couples the resulting modes globally before scattering them back to the spatial domain. This design is resolution-agnostic, scales near-linearly with the number of samples for a fixed modal budget, and naturally applies to irregular geometries and 3D domains. We separately evaluate Gaussian-basis reconstruction and operator prediction. Across standard PDE benchmarks and real reanalysis datasets, GPO achieves competitive accuracy against neural-operator and Transformer baselines, ranking best on most tasks and close to the best on the remainder. The learned particles and modal couplings further provide representation-level interpretability, positioning GPO as a useful intermediate representation rather than a one-to-one decomposition of physical structures. The code is available at https://github.com/lizhihao2022/GPO.

## 1. Introduction

Fluid-governed PDEs (Wazwaz, 2002; Gurtin, 1981) underpin many real-world systems, from numerical weather prediction and climate reanalysis to ocean circulation and engineering aerodynamics (McKeown et al., 2020; Shlesinger et al., 1987). Classical solvers (finite element/volume and spectral methods) (Wazwaz, 2002; Johnson, 2012; Klaasen & Troy, 1984) deliver high fidelity but face persistent challenges: strongly multi-scale dynamics, mesh dependence and complex geometries, stiffness in time integration, and high computational cost for long rollouts. Neural operators (Li et al., 2021; Kovachki et al., 2023; Lu et al., 2021) emerged as data-driven maps between function spaces, enabling resolution-agnostic surrogates; more recently, Transformer-based operators (Cao, 2021; Hao et al., 2023) leverage attention to capture long-range interactions and achieve strong empirical performance on diverse PDE tasks. However, these models share two recurring limitations: *(i) limited representation-level transparency*—latent features and attention weights are typically opaque and not aligned with explicit geometric or modal structures; and *(ii) localization/frequency bias*—global self-attention tends to favor low-rank, low-frequency correlations, making sharp fronts, vortical filaments, and other high-frequency structures harder to capture, while naïvely scaling attention over $N$ spatial samples incurs $\mathcal{O}(N^2)$ cost (Li et al., 2025).

We advocate representing fluid fields with a learned *Gaussian (particle) basis* rather than a fixed grid, hand-picked spectra (Gupta et al., 2021; Li et al., 2026b), or a monolithic implicit network (Serrano et al., 2023; 2024). Gaussian atoms carry *explicit geometric parameters*—centers and (anisotropic) scales—which tend to align with coherent flow structures (vortices, filaments, fronts) in our visualizations, afford multi-scale locality, and are directly *visualizable* and *differentiable*. The basis is compact, supports irregular boundaries, and extends from 2D to 3D query sets via the same evaluation rule (Buhmann, 2000; Park & Sandberg, 1991). While prior neural representations often rely on global Fourier features, wavelets, or black-box INRs, *learning a particleized Gaussian basis as the primary state of the field* has been relatively under-explored in the operator-learning setting. Concretely, a field is approxi-

---

[1]The Hong Kong University of Science and Technology (Guangzhou), Guangzhou, China [2]The Hong Kong University of Science and Technology, Hong Kong SAR, China. Correspondence to: Zhihao Li <zli416@connect.hkust-gz.edu.cn>, Wei Wang <weiwcs@ust.hk>.

*Proceedings of the 43rd International Conference on Machine Learning*, Seoul, South Korea. PMLR 306, 2026. Copyright 2026 by the author(s).

mated by weighted Gaussians with $\mu_i$ (centers), $\sigma_i$ (scales, possibly anisotropic), and $w_i$ (mixture weights) learned from data; evaluating these atoms at query locations yields a compact coefficient vector that serves as the latent state (basis). We emphasize that this provides *representation-level* interpretability—the geometric parameters and basis coefficients can be inspected—but does not, on its own, constitute a one-to-one physical decomposition.

We present a resolution-agnostic neural operator, the *Gaussian Particle Operator* (GPO), that learns a Gaussian-particle basis for fields and couples it with a *Petrov–Galerkin Gaussian Attention* layer, enabling basis-to-basis modeling with near-linear complexity in $N$ on 2D, 3D, and irregular domains. Our contributions are:

(1) **Gaussian Particle Representation.** An encoder learns per-site Gaussians $(\mu, \sigma, w)$; evaluating at arbitrary queries yields a visualizable basis $Z$ with explicit geometric parameters, applicable to regular and irregular query sets and to 3D.

(2) **PG Gaussian Attention.** Learned *Gaussian modal windows* perform a PG-style measurement ($N \to G$), a $G \times G$ attention parameterizes the modal coupling kernel, and the result is scattered back ($G \to N$), yielding an operator whose intermediate quantities (windows, modal tokens, couplings) are inspectable.

(3) **Efficiency.** With a small modal budget $G \ll N$, spatial transfers scale $\mathcal{O}(N)$ and modal attention is independent of $N$, supporting multi-step operator stacking with near-linear growth in resolution.

(4) **Empirical evaluation.** On synthetic PDE benchmarks (NS2D/NS3D, additional PDEBench tasks) and real reanalysis datasets (ERA5, CARRA), GPO is competitive with strong neural-operator and Transformer baselines; gains over the strongest baseline are sometimes moderate and task-dependent, while GPO additionally provides representation-level diagnostics (particle and modal visualizations) and improved spectral retention and rollout stability on the cases we examine.

**Conflict of Interest Disclosure.** The authors declare no financial conflicts of interest related to this work. None of the evaluation datasets (NS2D, PDEBench, ERA5, CARRA) or baseline implementations were developed by the authors' employers in connection with this submission.

## 2. Related Work

**Neural operators.** Neural operators learn mappings between function spaces. Representative designs include FNO and its variants (Li et al., 2021; Kovachki et al., 2023), DeepONet (Lu et al., 2021), multiresolution bases

(Li et al., 2020b; Gupta et al., 2021; He et al., 2024; Li et al., 2026b), graph or kernelized message passing (Li et al., 2025; 2020a), and recent diffusion-based super-resolution (Li et al., 2026a). Transformer operators replace hand-crafted kernels with attention—e.g., Galerkin Transformers (Cao, 2021), GNOT (Hao et al., 2023), Transolver (Wu et al., 2024), and orthogonal-attention operators (Xiao et al., 2024). These methods are predominantly data-driven: their latent state is typically opaque, attention weights are not anchored to explicit trial/test functions, and global self-attention often exhibits a low-frequency bias on sharp or localized structures.

**Gaussian and particle-style representations.** Anisotropic Gaussians have long been used as a flexible local basis, with RBF approximation guarantees on $\mathbb{R}^d$ (Park & Sandberg, 1991; Buhmann, 2000). The graphics community recently adopted closely related primitives: NeRF encodes scenes by an implicit MLP (Mildenhall et al., 2020), and 3D Gaussian Splatting (3DGS) renders scenes as an explicit set of anisotropic Gaussians with learned positions, covariances, and weights (Kerbl et al., 2023). Our Gaussian particle basis shares the explicit-state spirit of 3DGS but differs in three ways: (i) the basis serves as the *trial space of an operator*, not a static-scene renderer; (ii) particle parameters $(\mu, \sigma, w)$ are predicted by an encoder *per input field*, so the basis is input-adaptive rather than fitted per scene; and (iii) cross-particle interaction is parameterized in a learned modal space via PG-style attention, rather than by image-space compositing. Compared with INR/neural-field operator approaches (Serrano et al., 2023; 2024), the per-site Gaussian parameters provide an inspectable geometric state; compared with multiresolution operators with *fixed* bases, ours is a *learned* anisotropic basis whose locations and scales adapt to the input. A fuller discussion is in App. A.

## 3. Methodology

**Reconstruction vs. prediction.** Throughout the paper we distinguish two settings that share the same Gaussian basis. (i) *Reconstruction*: an encoder produces Gaussian particles from an observed field $a(\mathbf{x})$ and a decoder maps the resulting basis $Z$ back to $a$ at the queried locations; this is used to assess how faithfully the learned basis represents fields and to visualize the particles (Sec. 3.1, "Reconstruction" experiments). (ii) *Prediction*: the encoder produces particles from an input field $a$, the PG Gaussian Attention operator (Sec. 3.2) acts on the basis, and the decoder maps the updated basis to the target field $u$ (one-step prediction) or to the next field in a rollout (multi-step prediction). The reconstruction and prediction settings are reported separately, with metrics indicated explicitly in each table and figure caption.

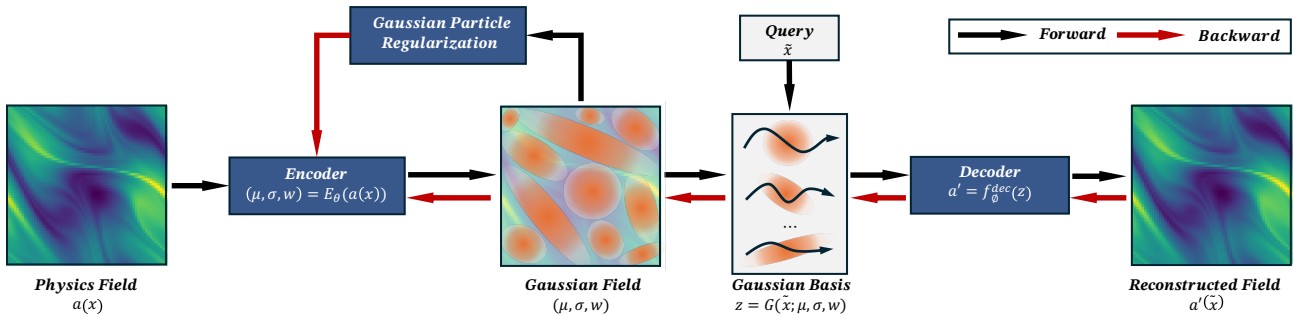

*Figure 1.* **Overview of the Gaussian basis representation (reconstruction setting).** Given the input field $a(\mathbf{x})$, an encoder $E_\theta$ produces $G$ Gaussian components per spatial location (mean $\mu$, scale $\sigma$, and mixture weight $w$). These define a Gaussian field that is evaluated at queries to form the basis $z$, which is decoded by $f_\phi^{\text{dec}}$ to reconstruct the input field $\hat{a}(\mathbf{x})$. In the prediction setting (Fig. 2), a modal operator $\mathcal{O}_\psi$ acts on $z$ before decoding to the target field $\hat{u}(\mathbf{x})$.

## 3.1. Gaussian Basis Representation

**From physics field to Gaussian field and basis.** We represent a spatial field $a : \Omega \subset \mathbb{R}^d \to \mathbb{R}^{d_a}$ by a set of *Gaussian particles* placed at each sample location $\mathbf{x}_j$. Each particle is parameterized by a center $\mu_{j,i} \in \mathbb{R}^d$, axis-aligned scale $\sigma_{j,i} \in \mathbb{R}^d$, and mixture weight $w_{j,i} \in [0,1]$ with $\sum_{i=1}^{G} w_{j,i} = 1$. The associated (unnormalized) kernel is

$$G(\tilde{\mathbf{x}}; \mu_{j,i}, \sigma_{j,i}) = \exp\left( -\frac{1}{2} \left\| (\tilde{\mathbf{x}} - \mu_{j,i})/\sigma_{j,i} \right\|_2^2 \right). \quad (1)$$

Evaluating these kernels at query $\tilde{\mathbf{x}}_j$ yields the *Gaussian basis* coefficients

$$z_{j,i} = w_{j,i}\, G(\tilde{\mathbf{x}}_j; \mu_{j,i}, \sigma_{j,i}), \quad (2)$$

where $\mathbf{z}_j = [z_{j,1}, \ldots, z_{j,G}]^\top \in \mathbb{R}^G$. The Gaussian field acts as a mollified, locally supported expansion of $a(\cdot)$; the coefficients in Eq. 2 can be viewed as localized averages of $a$ under data-adaptive windows $(\mu, \sigma)$, while $w$ distributes mass among overlapping particles. This basis is resolution-agnostic and applies to irregular query sets.

**Encoder.** Given samples $\{(x_j, a_j)\}_{j=1}^N$, the encoder $E_\theta$ injects geometry by embedding coordinates and fusing them with field features. Concretely, we form $\eta_j = [\, a_j,\ \gamma(x_j)\,]$, where $\gamma(\cdot)$ is a fixed Fourier-feature map (or a small MLP), and compute

$$\phi_j = \text{ReLU}(W_{\text{in}} \eta_j), \quad (3)$$
$$\mu_j = W_\mu\, \text{ReLU}(U_\mu \phi_j), \quad (4)$$
$$\sigma_j = \text{Softplus}\big(W_\sigma\, \text{ReLU}(U_\sigma \phi_j)\big), \quad (5)$$
$$w_j = \text{Softmax}\big(W_w\, \text{ReLU}(U_w \phi_j)\big), \quad (6)$$

reshaped as $\mu_{j,i}, \sigma_{j,i} \in \mathbb{R}^d$ and $w_{j,i} \in [0,1]$ with $\sum_i w_{j,i} = 1$. We use $\gamma(x)$ as a fixed positional encoding (Fourier features): $\gamma(x) = [\sin(2\pi Bx), \cos(2\pi Bx)]$, where $B \in \mathbb{R}^{m \times d}$ is sampled once from $\mathcal{N}(0, \sigma_B^2)$ and kept fixed; the resulting coordinate embedding has dimension $2m$.

**Gaussian basis evaluation.** With $(\mu, \sigma, w)$ predicted by $E_\theta$, the weighted Gaussian evaluation Eq. 2 produces the per-site latent vector $\mathbf{z}_j$. *Geometrically*, $\mu$ encodes particle locations, $\sigma$ controls receptive-field sizes (anisotropy along axes), and $w$ balances overlapping contributions; these parameters provide a representation-level view of the basis but are not, on their own, claimed to coincide with physical degrees of freedom. *Computationally*, the map $(\mu, \sigma, w, \tilde{\mathbf{x}}) \mapsto \mathbf{z}$ is local and embarrassingly parallel.

**Decoder.** A lightweight MLP head $f_\phi^{\text{dec}} : \mathbb{R}^G \to \mathbb{R}^{c_{\text{out}}}$ regresses from $\mathbf{z}_j$ to the field value at the query:

$$\hat{a}(\tilde{\mathbf{x}}_j) = f_\phi^{\text{dec}}(\mathbf{z}_j). \quad (7)$$

In practice, we use a two-layer perceptron with ReLU.

**Gaussian particle regularization.** Because the constraints act on the *particle parameters* produced by the encoder, we impose them at the Gaussian-field level (conceptually tied to $E_\theta$ but applied after parameter prediction):

$$\mathcal{L}_\mu = \frac{1}{N} \sum_{j=1}^N \left\| \sum_{i=1}^G w_{j,i} \mu_{j,i} - \mathbf{x}_j \right\|_2^2, \quad (8)$$

$$\mathcal{L}_\sigma = \frac{1}{NGd} \sum_{j,i,\ell} \Big[ [\sigma_{j,i,\ell} - \sigma_{\max}]_+ + [\sigma_{\min} - \sigma_{j,i,\ell}]_+ \Big], \quad (9)$$

which keep particle centers near their host coordinates (a representation-level prior), avoid degenerate particles, and discourage overly peaky mixtures.

**Overview pipeline.** Eqs. 2–7 define the Gaussian-field training pipeline, and the complete forward/backward diagram is in Fig. 1. We minimize the reconstruction loss together with $\mathcal{L}_\mu, \mathcal{L}_\sigma$.

**Approximation capacity of the Gaussian basis.** We record a standard density result:

**Lemma 3.1** (Density of Gaussian mixtures). *Let $\Omega \subset \mathbb{R}^d$ be compact. Finite linear combinations of (possibly anisotropic) Gaussian kernels are dense in $C(\Omega)$ and in $L^r(\Omega)$ for $1 \le r < \infty$. Consequently, for any continuous scalar field $v$ and $\varepsilon > 0$, there exist $G$ and parameters $\{(\mu_i, \sigma_i, c_i)\}_{i=1}^G$ with $c_i \in \mathbb{R}$ such that*

$$\left\| v(\cdot) - \sum_{i=1}^G c_i \exp\left( -\tfrac{1}{2} \|(\cdot - \mu_i)/\sigma_i\|_2^2 \right) \right\|_\infty < \varepsilon.$$

*Vector-valued fields admit componentwise approximation.* (Proof in App. B.1)

### 3.2. Petrov–Galerkin Gaussian Attention

Sec. 3.1 established *local*, per-point Gaussian bases: at each location $j$, $G$ weighted coefficients $\mathbf{z}_j \in \mathbb{R}^G$ are derived from particles $(\mu, \sigma, w)$. To learn a resolution-agnostic operator on this basis, we draw on the Petrov–Galerkin (PG) perspective—in which a field is approximated in a trial space and residuals are enforced to be orthogonal to a (possibly different) test space—and move from the spatial grid to a modal space (Franca et al., 2006; Brooks & Hughes, 1982). Concretely, learned Gaussian modal windows first *measure* the field by aggregating information across locations, a global *mode coupling* operates on the $G$ Gaussian components, and the result is then *scattered back* to locations. This pipeline yields a basis-to-basis operator that is computationally efficient ($G \ll N$) and whose intermediate quantities (windows, modal tokens, attention) are inspectable at the representation level.

#### 3.2.1. FROM PETROV–GALERKIN PROJECTION TO A GAUSSIAN-BASIS OPERATOR

In Petrov–Galerkin, we expand the field in a *trial* space and evaluate it with a *test* space. Here, Gaussian particles form the trials, while *Gaussian modal windows* serve as discrete tests that pool local information into global modes.

**Trial functions (Gaussian basis).** Let the unnormalized Gaussian particle (anchored at location $j$, component $i$) be

$$\phi_{j,i}(\mathbf{x}) = \exp\left( -\tfrac{1}{2} \|(\mathbf{x} - \mu_{j,i})/\sigma_{j,i}\|_2^2 \right), \Sigma_{j,i} = \mathrm{diag}(\sigma_{j,i}^2). \tag{10}$$

With weighted evaluations (Sec. 3.1), each site provides $\mathbf{z}_j = [z_{j,1}, \ldots, z_{j,G}]^\top \in \mathbb{R}^G$, where $z_{j,i} = w_{j,i}\, \phi_{j,i}(\tilde{\mathbf{x}}_j)$.

**Test functions (Gaussian modal windows).** We distinguish two normalizations: (i) row-normalized assignments $\sum_g p_{j,g} = 1$ (used to map each location to modes), and (ii) column-normalized weights $\bar{p}_{j,g} = p_{j,g}/\sum_{j'} p_{j',g}$ so that

$\sum_j \bar{p}_{j,g} = 1$ (used for PG measurement).

$$\psi_g(x) \approx \sum_{j=1}^N \bar{p}_{j,g}\, \delta(x - \tilde{x}_j), \qquad \bar{p}_{j,g} = \frac{p_{j,g}}{\sum_{j'} p_{j',g}}. \tag{11}$$

Using a linear projection of coefficients $\mathbf{s}_j = \mathbf{z}_j W_z \in \mathbb{R}^D$, the PG *measurement* (test of the trial field) yields modal tokens

$$t_g = \sum_{j=1}^N \bar{p}_{j,g}\, s_j, \qquad s_j = z_j W_z. \tag{12}$$

**Modal coupling and scatter.** Let $\kappa : \{1, \ldots, G\}^2 \to \mathbb{R}^{D \times D}$ be a (learned) coupling kernel over modes. PG updates the modal state and scatters it back:

$$U_g = \sum_{g'=1}^G \kappa(g, g')\, t_{g'} \in \mathbb{R}^D, \tag{13}$$

$$\widetilde{\mathbf{z}}_j = \left( \sum_{g=1}^G p_{j,g}\, U_g \right) W_{\mathrm{out}} \in \mathbb{R}^G. \tag{14}$$

Stacking sites gives $\widetilde{Z} \in \mathbb{R}^{N \times G}$. Algebraically,

$$\boxed{\widetilde{Z} \approx A\,\mathcal{K}\, A^\top\, (Z\, W_z)\, W_{\mathrm{out}},} \tag{15}$$

where $A[j, g] = p_{j,g}$ and $\mathcal{K}[g, g']$ encodes modal coupling. Thus PG supplies the *structure*: test (measure) $\to$ couple $\to$ scatter.

#### 3.2.2. ATTENTION AS A PARAMETERIZATION OF THE PG OPERATOR

We now instantiate Eq. 15 with a multi-head attention layer that is global in *modal* space and local in the $N \leftrightarrow G$ transfers. Let $Z \in \mathbb{R}^{N \times G}$ and particle parameters $(\mu, \sigma, w) \in \mathbb{R}^{N \times G \times d} \times \mathbb{R}^{N \times G \times d} \times \mathbb{R}^{N \times G}$.

**Learned Gaussian modal windows.** For head $h$, form a per-site descriptor $\xi_j = [\, \mathbf{z}_j, \mathbf{w}_j, \mu_j, \sigma_j \,] \in \mathbb{R}^{G(2d+2)}$ and project to $h_j^{(h)} \in \mathbb{R}^D$. A softmax over modes produces windows

$$p_{j,g}^{(h)} = \mathrm{softmax}_g\big(W_p^{(h)} h_j^{(h)}\big), \tag{16}$$

which instantiate the PG test functions (Eq. 11) in discrete form. $W_p^{(h)} \in \mathbb{R}^{G \times D}$ is the (head-specific) linear projection that maps the local embedding $h_j^{(h)}$ at location $j$ to mode logits over the $G$ Gaussian modes.

**PG measurement $N \to G$.** Project coefficients $\mathbf{s}_j^{(h)} = \mathbf{z}_j W_z^{(h)}$ and compute tokens

$$t_g^{(h)} = \frac{\sum_j p_{j,g}^{(h)}\, \mathbf{s}_j^{(h)}}{\sum_j p_{j,g}^{(h)}} \in \mathbb{R}^D, T^{(h)} = [t_1^h, \ldots, t_G^h] \in \mathbb{R}^{G \times D}, \tag{17}$$

which matches the PG measurement in Eq. 12.

*Table 1.* **One-step prediction performance comparison with baselines on benchmarks.** Relative $L_2$ error on the held-out test set, computed after inverse normalization. Bold: best in column. Underlined: second-best. Gains over the strongest baseline are sometimes moderate (e.g., NS2D, PlanetSWE) and we report numbers rather than claim broad SOTA.

| MODEL | NS2D | NS3D | ERA5-TEMP | ERA5-WIND U | CARRA | AIRFOIL | TURBULENT | PLANETSWE |
|---|---|---|---|---|---|---|---|---|
| (GEO-)FNO | 3.24E-02 | 5.07E-01 | 7.09E-03 | 1.02E-01 | 3.50E-01 | 5.12E-03 | 4.82E-01 | 9.23E-02 |
| M2NO | 2.76E-02 | 4.65E-01 | 3.35E-03 | 7.15E-02 | 3.62E-01 | 1.98E-03 | 4.53E-01 | 8.96E-02 |
| CNO | 9.27E-02 | 8.12E-01 | 1.24E-02 | 1.86E-01 | 6.89E-01 | 4.65E-03 | 7.91E-01 | 1.57E-01 |
| LSM | 3.11E-02 | 3.80E-01 | 5.86E-03 | 8.23E-02 | 4.05E-01 | 5.37E-03 | 4.16E-01 | 9.58E-02 |
| AMG | 3.12E-02 | 4.32E-01 | 6.11E-03 | 8.79E-02 | 3.21E-01 | 3.09E-03 | 4.47E-01 | 8.14E-02 |
| GT | 8.81E-02 | 5.39E-01 | 5.44E-03 | 1.55E-01 | 3.73E-01 | 8.26E-03 | 5.41E-01 | 1.23E-01 |
| GNOT | 7.19E-01 | 1.01E+00 | 1.55E-02 | 3.49E-01 | 7.57E-01 | 1.53E-02 | 9.86E-01 | 2.14E-01 |
| TRANSOLVER | 3.76E-02 | 5.29E-01 | 4.18E-03 | 1.06E-01 | 3.76E-01 | 9.46E-03 | 5.33E-01 | 1.05E-01 |
| ONO | 4.26E-02 | 8.83E-01 | 1.45E-02 | 3.49E-01 | 7.25E-01 | 5.26E-03 | 8.27E-01 | 1.69E-01 |
| LNO | 4.81E-02 | 3.68E-01 | 7.32E-03 | 1.31E-01 | 4.36E-01 | 4.39E-03 | 5.69E-01 | 9.12E-02 |
| **GPO** | **3.02E-02** | **3.44E-01** | **2.26E-03** | **6.68E-02** | **2.97E-01** | **1.12E-03** | **3.95E-01** | **8.01E-02** |

**Global modal coupling ($G \times G$ attention).** Scaled dot-product attention parameterizes the kernel $\mathcal{K}$:

$$Q^{(h)} = T^{(h)} W_Q^{(h)}, \quad K^{(h)} = T^{(h)} W_K^{(h)}, \quad V^{(h)} = T^{(h)} W_V^{(h)},$$
$$\tag{18}$$

$$\alpha^{(h)} = \mathrm{softmax}\left(\frac{Q^{(h)} K^{(h)\top}}{\sqrt{D}}\right), \tag{19}$$

$$\widetilde{T}^{(h)} = \alpha^{(h)} V^{(h)} \ \in \ \mathbb{R}^{G \times D}. \tag{20}$$

Here $\alpha^{(h)}(g, g')$ plays the role of a data-driven modal coupling $\kappa(g, g')$.

**Scatter $G \to N$ and readout.** Using the same windows, scatter the coupled modes back and read out to $G$ coefficients:

$$y_j^{(h)} = \sum_g p_{j,g}^{(h)} U_g^{(h)} \in \mathbb{R}^D, \tag{21}$$

$$\widetilde{\mathbf{z}}_j = \left(\Big\|_{h=1}^{H} y_j^{(h)}\right) W_{\mathrm{out}} \in \mathbb{R}^G, \tag{22}$$

$$\widetilde{Z} = [\widetilde{\mathbf{z}}_1^\top; \ldots; \widetilde{\mathbf{z}}_N^\top]. \tag{23}$$

To stabilize training and preserve the per-site total mass (row-wise $\ell_1$ sum), we first take a convex residual update with a mixing coefficient $\lambda \in [0, 1]$ and then renormalize each row:

$$\widehat{Z} = (1 - \lambda) Z + \lambda \widetilde{Z}, \tag{24}$$

$$Z'_{j,:} = \frac{\sum_{g=1}^{G} Z_{j,g}}{\sum_{g=1}^{G} \widehat{Z}_{j,g} + \varepsilon} \widehat{Z}_{j,:}, \qquad j = 1, \ldots, N, \tag{25}$$

where $\varepsilon > 0$ avoids division by zero. Eq. 24 provides a conservative blend between the old and updated coefficients, while Eq. 25 rescales each site's coefficients so that $\sum_g Z'_{j,g} = \sum_g Z_{j,g}$.

*Complexity.* Per head, forming the window logits requires projecting $\xi_j \in \mathbb{R}^{G(2d+2)}$ to $\mathbb{R}^D$, costing $\mathcal{O}(BHN \cdot G(2d + 2) \cdot D)$. The two transfers $N \leftrightarrow G$ cost $\mathcal{O}(BHNGD)$, and modal attention costs $\mathcal{O}(BHG^2D)$ (independent of $N$). Overall, for fixed $G$ the dominant term is linear in $N$, with explicit dependence on spatial dimension $d$.

**Expressivity of the modal operator.** We formalize that PG Gaussian Attention can approximate a broad class of continuous operators:

**Theorem 3.2** (Universal approximation in modal form)**.** *Let* $\mathcal{T} : L^p(\Omega; \mathbb{R}^{c_{\mathrm{in}}}) \to L^q(\Omega; \mathbb{R}^{c_{\mathrm{out}}})$ *be continuous on bounded sets and admit either a Mercer/Hilbert–Schmidt kernel* $K(\mathbf{x}, \mathbf{x}')$ *or a low-rank factorization* $\mathcal{T} \approx \Phi(\cdot) \mathcal{K} \Phi(\cdot)^\top$ *with continuous* $\Phi : \Omega \to \mathbb{R}^m$*. Then, for any* $\varepsilon > 0$*, there exist a modal budget* $G$ *and parameters* $\Theta$ *of our encoder, Gaussian modal windows, PG Gaussian Attention, and decoder such that* $\|\mathcal{G}_\Theta - \mathcal{T}\| < \varepsilon$ *(operator norm on bounded subsets).* (Proof in App. B.2)

### 3.3. Gaussian Particle Operator: Overall Framework

#### 3.3.1. NEURAL OPERATOR FORMULATION

Let $\Omega \subset \mathbb{R}^d$ be the domain, $a : \Omega \to \mathbb{R}^{c_{\mathrm{in}}}$ the input field, and $u : \Omega \to \mathbb{R}^{c_{\mathrm{out}}}$ the target field. We model the map $a \mapsto u$ by a neural operator

$$\mathcal{G}_\Theta = f_\phi^{\mathrm{dec}} \circ \left(\mathcal{O}_\psi\right)^{\circ n} \circ \mathcal{Z}\big(\cdot \,; \Pi_\theta(\cdot)\big) \circ E_\theta, \tag{26}$$

where:

- $E_\theta$ (encoder) predicts Gaussian particles $\Pi_\theta(a) = \big(\mu_\theta, \sigma_\theta, w_\theta\big)$ on the *context* locations $\{x_j\}_{j=1}^N$ (using both $a_j$ and $x_j$);

- $\mathcal{Z}(\cdot; \Pi)$ evaluates the *Gaussian basis* and returns per-location, $G$-dimensional coefficients $Z \in \mathbb{R}^{N \times G}$ with

$$z_{j,i} = w_{j,i} \exp\left(-\tfrac{1}{2}\big\|(\mathbf{x}_j - \mu_{j,i})/\sigma_{j,i}\big\|_2^2\right); \tag{27}$$

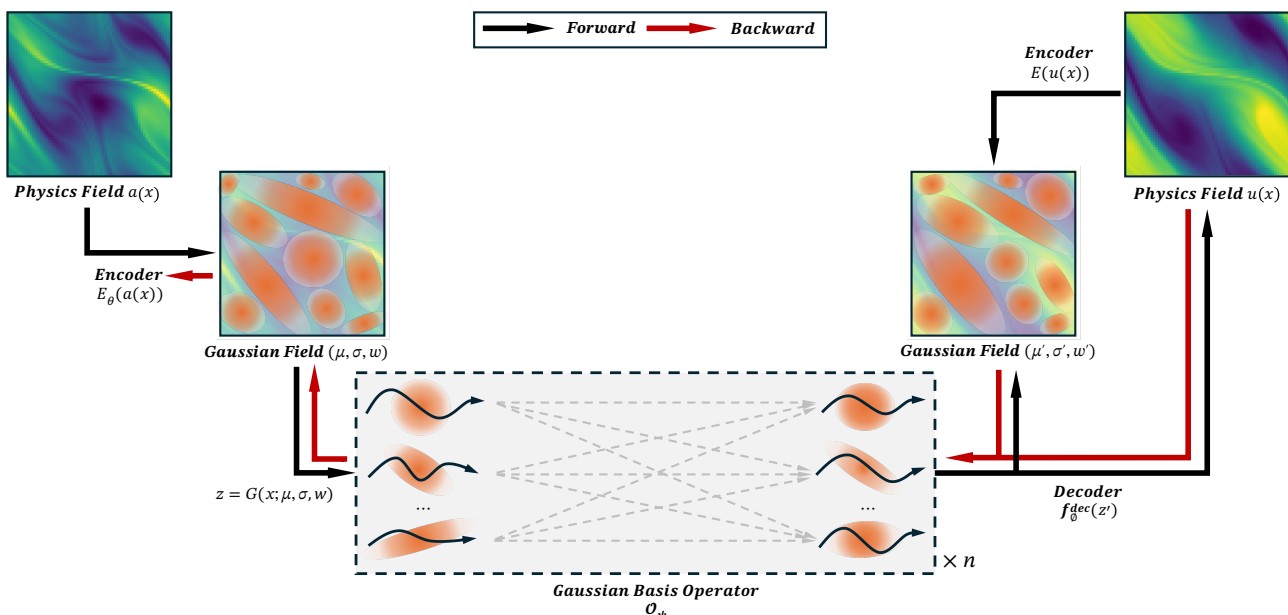

*Figure 2.* **Architecture of the Gaussian Particle Operator (GPO) for prediction.** The pipeline encodes the input field $a(\mathbf{x})$ into a Gaussian field $(\mu, \sigma, w)$, evaluates a basis $Z$, applies the modal operator $\mathcal{O}_\psi$, and decodes to the predicted target $\hat{u}(\mathbf{x})$; black arrows denote forward computation, red arrows denote gradients. The reconstruction setting bypasses $\mathcal{O}_\psi$ and decodes $Z$ back to $\hat{a}(\mathbf{x})$.

- $\mathcal{O}_\psi$ is the *Gaussian-basis operator* (Sec. 3.2) acting on $Z$ and parameterized by PG Gaussian Attention; it can be applied $n$ times:

$$Z^{(0)} = Z, \tag{28}$$

$$Z^{(k+1)} = \mathcal{O}_\psi\big(Z^{(k)}; \Pi_\theta(a)\big), \quad k = 0, \ldots, n-1. \tag{29}$$

- $f_\phi^{\text{dec}}$ (decoder) maps the updated basis to the output field values: $\hat{u}(\mathbf{x}_j) = f_\phi^{\text{dec}}(Z_{j,:}^{(n)})$.

The construction is resolution-agnostic: for any query set $\{\mathbf{x}_j\}$—on 2D/3D grids or irregular meshes—one simply recomputes Eq. 27 and reuses the same $\mathcal{O}_\psi$ and $f_\phi^{\text{dec}}$.

### 3.3.2. PIPELINE OVERVIEW

As shown in Fig. 2, given an input field $a(\mathbf{x})$, the encoder $E_\theta$ produces per-site Gaussian particles $\Pi_\theta(a) = (\mu, \sigma, w)$, i.e., a *Gaussian field*. At query locations $\{\mathbf{x}_j\}_{j=1}^N$, we then evaluate the Gaussian basis by Eq. 27 to obtain $Z \in \mathbb{R}^{N \times G}$. The Gaussian-basis operator $\mathcal{O}_\psi$ acts on $Z$ in modal space and can be applied for $n$ stages as in Eq. 28 to capture multi-step coupling, yielding $Z^{(n)}$. Finally, the decoder $f_\phi^{\text{dec}}$ maps $Z_{j,:}^{(n)}$ to $\hat{u}(\mathbf{x}_j)$. During training, the target $u(\mathbf{x})$ may also be encoded by $E_\theta$ to provide an auxiliary Gaussian-field supervision signal.

## 4. Experiments

**Benchmarks.** We evaluate on synthetic Navier–Stokes surrogates and real reanalyses to span 2D→3D and regular→irregular domains. **NS2D** (Kovachki et al., 2023) is an incompressible periodic box sampled on $64 \times 64$; **NS3D** (Takamoto et al., 2022) extends to a periodic cube on $64^3$. **ERA5** (Hersbach et al., 2023) uses one month on the $0.25°$ global grid ($721 \times 1440$), with variables 2 m temperature ($t$) and 10 m zonal wind ($u$). **CARRA** (Schyberg et al., 2020) uses one Arctic month on its native regional grid ($989 \times 789$) with an irregular land/sea/ice mask, variables 10 m meridional wind ($v_{10}$) and surface pressure ($sp$). We additionally include PDEBench surrogates (Airfoil, Turbulent, PlanetSWE) to broaden the comparison. For prediction tasks, we train one-step operators and assess multi-step rollouts on native grids, applying latitude weighting on ERA5 and CARRA.

**Reconstruction vs. prediction.** Two settings are reported throughout this section. Reconstruction measures how well the learned Gaussian basis encodes and decodes a given field (Sec. 3.1); prediction measures how well the full GPO maps an input field to a target field (one-step) or rolls out future states (multi-step). Tabs. 1 and 7 and Figs. 3–4 concern prediction. Figs. 5 and 7 concern reconstruction. The two settings share the encoder/decoder; only prediction additionally invokes the modal operator $\mathcal{O}_\psi$.

**Baselines.** We compare with physics-inspired neural operators—**FNO** (Li et al., 2021), **LSM** (Wu et al., 2023), **M2NO** (Li et al., 2026b), and **MgNO/AMG** (He et al., 2024)—and Transformer-based operators—**Galerkin Transformer** (Cao, 2021), **GNOT** (Hao et al., 2023), **ONO** (Xiao et al., 2024), and **Transolver** (Wu et al., 2024). Additional spectral baselines (CNO, LNO) reported in Tab. 1 follow the implementations from the Neural-Solver-Library. All baselines use identical data splits, losses, and rollout protocols. For a capacity-fair check, we further compare against scaled-up FNO variants matched to GPO's parameter budget (App. D.4).

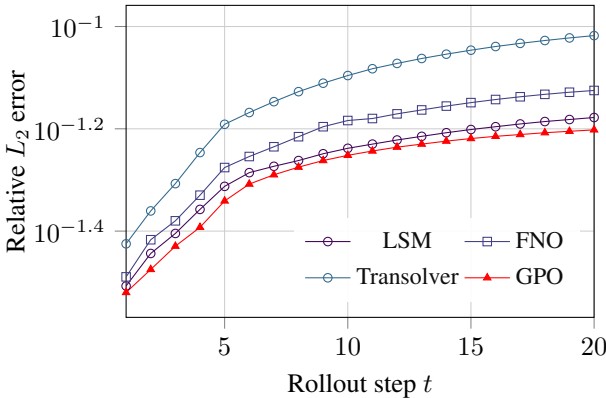

*Figure 3.* **Multi-step prediction (rollout) on NS2D**: relative $L_2$ error versus rollout horizon. GPO shows comparatively stable error growth over the displayed horizon.

**Implementations.** We evaluate all models using the relative $L_2$ error on held-out sets. Inputs/targets are *normalized per variable* using training statistics; models are trained on the normalized data, and *all metrics are computed after inverse normalization*. Training uses AdamW (Loshchilov & Hutter, 2019) with an initial learning rate of $10^{-3}$ and a StepLR scheduler. All experiments are run on a single NVIDIA RTX 4090 GPU. Per-dataset and per-baseline hyperparameters are in App. C.

### 4.1. Prediction Benchmark Performance

Tab. 1 reports one-step prediction $L_2$ errors across Navier–Stokes surrogates, PDEBench tasks, and real reanalyses.

**(i) Synthetic, regular grids (NS2D / NS3D).** GPO is the lowest-error entry in our setup ($3.02 \times 10^{-2}$ and $3.44 \times 10^{-1}$), but its margin over the next strongest entries (M2NO on NS2D, LNO/LSM on NS3D) is moderate—these tasks favor spectral inductive biases. A capacity-fair FNO of comparable parameter budget (App. D.4) closes part of the gap but not all of it, which we attribute to the basis-to-basis design rather than to capacity alone.

**(ii) Real reanalyses (ERA5 / CARRA).** The gap to spectral and attention baselines is larger: GPO is best on *ERA5-temp* ($2.26 \times 10^{-3}$), *ERA5-wind u* ($6.68 \times 10^{-2}$), and *CARRA* ($2.97 \times 10^{-1}$). This is consistent with the mesh-/mask-agnostic Gaussian basis: the same encoder/decoder applies on the $0.25°$ ERA5 grid (with latitude weighting) and on the masked CARRA grid, without changes to the operator.

**(iii) PDEBench (Airfoil, Turbulent, PlanetSWE).** GPO is again lowest in our runs, but on Airfoil and PlanetSWE the margin over the next-best baseline is small ($\sim$10–30% relative). We therefore do not claim broad SOTA; the results are best read as evidence that the Gaussian-basis design is competitive across diverse settings rather than uniformly dominant. Multi-step rollout stability and predicted spectra on NS2D are shown in Figs. 3 and 4.

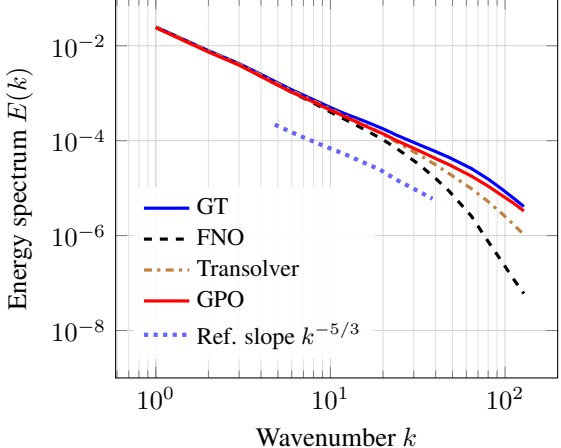

*Figure 4.* **NS2D predicted energy spectrum (log–log).** FNO exhibits earlier high-$k$ energy decay, Transolver is intermediate, and GPO more closely tracks the ground-truth spectrum on the cases shown.

### 4.2. Representation-Level Visualizations

**Reconstruction of input fields.** Before assessing the prediction operator, we verify that the learned Gaussian basis is faithful and inspectable in the *reconstruction* setting. We train the encoder–decoder (Sec. 3.1) to map an input field to itself through the Gaussian basis, using the weighted Gaussian evaluation and the particle regularizers. As shown in Fig. 5 for in-distribution (ID) and out-of-distribution (OOD) cases: (i) particles tend to concentrate along coherent flow structures, and their anisotropic scales align with local intensity gradients—we describe this as an empirical alignment rather than a one-to-one physical decomposition; (ii) reconstruction error maps are predominantly localized near sharp gradients or subgrid filaments; (iii) under OOD shifts, the representation remains stable: particle geometry and weights adapt to novel patterns and reconstruction errors increase modestly. Quantities here describe reconstruction;

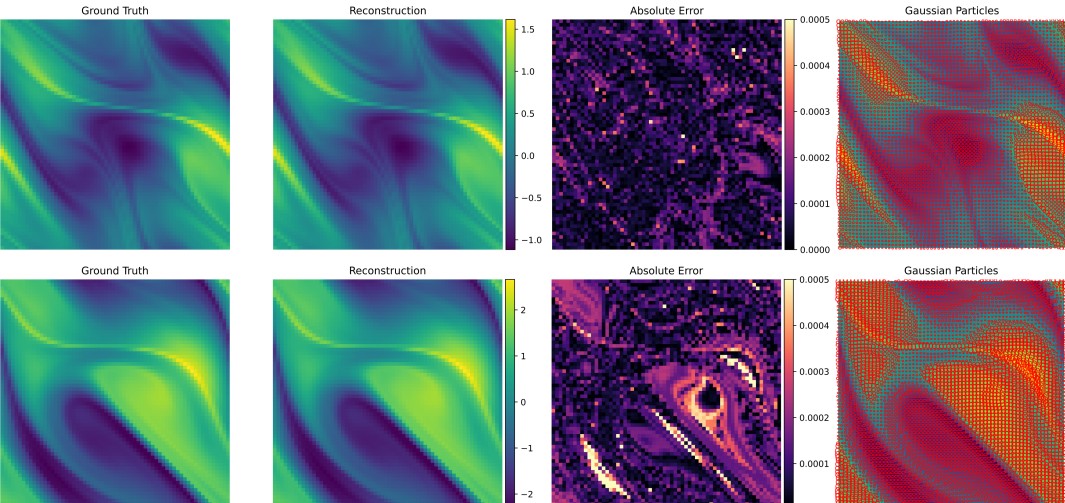

*Figure 5.* **Reconstruction setting:** representation-level visualization on an **in-distribution sample (top)** and an **out-of-distribution sample (bottom)**. Left to right: ground-truth field, reconstruction from the Gaussian basis, absolute reconstruction error, and learned Gaussian particles overlaid on the input (ellipses: center $\mu$, axes $\propto \sigma$, color/size $\propto w$). Reconstruction reflects how well the basis encodes/decodes a given field; prediction errors are reported separately in Tab. 1.

predictions are reported separately in Tab. 1. Additional reconstruction visualizations on ERA5 are in App. E.

**Layer-wise dynamics in the prediction operator.** In the prediction setting, the Gaussian Particle Operator (Sec. 3.2) is applied for $n$ stages; the encoder-fixed particles $(\mu, \sigma)$ define the trial atoms, while PG Gaussian Attention updates the per-site coefficients $\mathbf{z}_j^{(k)} \in \mathbb{R}^G$. We visualize the particle field at each stage by overlaying particle footprints with color proportional to the local activation $A^{(k)}(\mathbf{x}_j) = \sum_g z_{j,g}^{(k)}$ (Fig. 6). Layer-wise changes admit suggestive but *qualitative* readings—diffusion-like mixing among nearby modes, advection-like off-diagonal transfers, and cross-scale activation growth/decay—and should not be read as evidence that individual particles correspond to distinct physical modes.

### 4.3. Model Analysis

**Component ablations.** Tab. 3 isolates the two design choices on NS2D (one-step prediction). Removing the Gaussian Field and using a plain MLP encoder/decoder degrades $L_2$ from $3.90 \times 10^{-2}$ to $7.44 \times 10^{-2}$; removing the PG operator while keeping the Gaussian Field is worse still ($8.57 \times 10^{-2}$). Even a single Gaussian per site (num_gaussian=1) already improves over the plain MLP ($6.28 \times 10^{-2}$), indicating that the particleized Gaussian evaluation is itself a useful inductive bias. Increasing the modal budget $G$ from 16 to 64 reduces error from $4.21 \times 10^{-2}$ to $3.84 \times 10^{-2}$ with diminishing returns, so we use $G \in \{16, 32\}$ in all main-table runs.

**Kernel choice.** Under an identical encoder/operator/decoder pipeline, replacing the anisotropic Gaussian basis with a Laplacian, multiquadric RBF, or inverse-quadric RBF gives $L_2$ errors of $5.18 \times 10^{-2}$, $5.27 \times 10^{-2}$, and $4.86 \times 10^{-2}$ respectively (Tab. 4), all above the Gaussian's $3.90 \times 10^{-2}$. The advantage of the Gaussian basis on flow tasks is consistent with its anisotropic, locally smooth support.

**Regularizer sensitivity.** The center regularizer $\mathcal{L}_\mu$ controls how tightly particle centers track their host coordinates (Tab. 5). Removing $\mathcal{L}_\mu$ inflates the center error to $1.87 \times 10^{-2}$ (vs. $4.35 \times 10^{-3}$ at the default weight) and also degrades scale regularity; relaxing the weight to $0.1 \times$ marginally improves $L_2$ but at the cost of geometric regularity, while a strong $5 \times$ weight gives the tightest centers but a small $L_2$ increase. The default weight is what we use elsewhere.

**Capacity-fair FNO.** Scaling FNO on the Turbulent task either in width (channel $128 \to 256$, FC $256 \to 512$) or in the number of retained modes (per-direction modes $[15, 12, 9, 9, 9] \to [21, 18, 15, 15, 15]$) only partially closes the gap to GPO: $4.82 \times 10^{-1} \to 4.78 \times 10^{-1}$ (wider) and $\to 4.63 \times 10^{-1}$ (more modes), still above GPO's $3.95 \times 10^{-1}$ at a comparable parameter count of $\sim 1.6\,\text{M}$ (Tab. 6). We read this as evidence that GPO's gain on multi-scale tasks is driven by the basis-to-basis design rather than by parameter count.

**Complexity.** Tab. 7 shows that GPO retains a low memory footprint (2,313 MiB) and competitive runtime (44.66 / 1.67

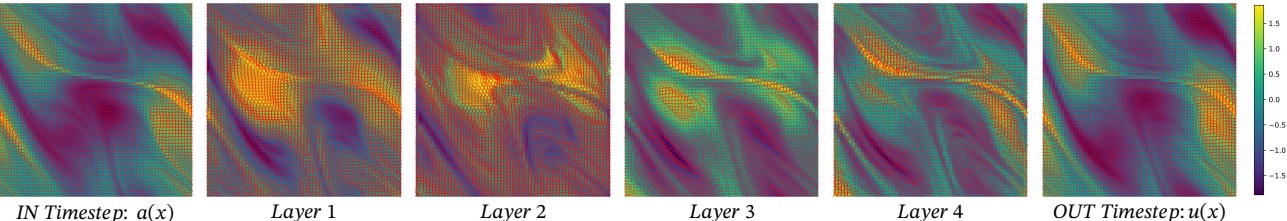

*IN Timestep: $a(x)$*     *Layer 1*     *Layer 2*     *Layer 3*     *Layer 4*     *OUT Timestep: $u(x)$*

*Figure 6.* **Layer-wise evolution of the Gaussian particle field (prediction setting).** Particle activations after successive PG Gaussian Attention layers; we read these as qualitative analogies (diffusion-like / advection-like / cross-scale exchange) rather than a one-to-one physical decomposition.

s/epoch train/inference) at $\sim$6 MB of parameters, scaling near-linearly with $N$. Per-baseline numbers, training curves, and additional analyses are in App. D.

## 5. Conclusion

We introduced the *Gaussian Particle Operator* (GPO): a resolution-agnostic neural operator that represents fields by a learned Gaussian (particle) basis and performs basis-to-basis coupling via Petrov–Galerkin Gaussian Attention. The design exposes its intermediate objects—particles $(\mu, \sigma, w)$, modal windows, and inter-modal couplings—as visualizable quantities, providing representation-level interpretability. In our experiments, GPO is competitive with strong neural-operator and Transformer baselines on synthetic NS2D/NS3D, PDEBench tasks, and large real-world datasets (ERA5, CARRA), and shows improved spectral retention and rollout stability on the cases we examined. Gains over the strongest baseline are sometimes moderate and task-dependent; we therefore frame the contribution as a basis-to-basis design with explicit intermediate representations, complementary to the empirical improvements.

**Limitations and future work.** Performance depends on the modal budget $G$ and head width; although $N \leftrightarrow G$ transfers are linear in $N$, the memory and compute of $NG$ windows can still be a bottleneck at extreme resolutions. Future work includes adaptive or hierarchical particles (multi-scale $G$), sparse/modal pruning and routing, structured/low-rank attention in $G$-space, and optimized implementations (mixed precision, kernel fusion). The current training is primarily data-driven with lightweight particle regularization; it does not *guarantee* physical invariants (e.g., mass/energy conservation) or constraints (e.g., divergence-free flow, boundary conditions), and the particles should not be read as a one-to-one decomposition of physical structures. We plan to couple the Gaussian basis more tightly with physics so that particle parameters can be more directly aligned with physically evolving features—a step toward mechanistic, rather than merely representation-level, interpretability.

## Impact Statement

This work develops a neural operator for surrogate modeling of partial differential equations, with a focus on fluid dynamics, weather, and climate-related fields. Faster and more flexible surrogates can support scientific machine learning by accelerating simulation-heavy workflows (e.g., ensemble exploration, hyperparameter studies, data assimilation prototyping) and by reducing the energy cost of repeated solves. The explicit Gaussian basis offers representation-level diagnostics that may help researchers inspect intermediate states. Inaccurate surrogate predictions can mislead downstream scientific or engineering decisions, especially in safety-critical settings such as operational weather forecasting, climate impact assessment, or aerospace design; out-of-distribution inputs may produce confident but incorrect outputs. We therefore recommend that GPO outputs be validated against high-fidelity solvers or observations before being used in any deployed setting, and that the model not be used as the sole basis for safety-critical decisions. We do not foresee unique societal risks beyond those generally associated with PDE surrogate models.

## Acknowledgements

The work of Wei Wang is supported by the Advanced Materials—National Science and Technology Major Project (Grant No. 2025ZD0620100), HKUST(GZ)-IEIP-RoP (G01RF000256), the National Key R&D Program of China (No. 2024YFA1012700), and the Guangdong Provincial Key Laboratory of Integrated Communication, Sensing and Computation for Ubiquitous Internet of Things (No. 2023B1212010007). The work of Zhilu Lai is supported by the Guangdong Provincial Fund—Special Innovation Project (2024KTSCX038) and the Research Grants Council of Hong Kong through the Research Impact Fund (R5006-23).

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

## A. Related Work (Extended)

**Neural operators.** Classical neural-operator methods aim to learn mappings between function spaces directly from data, typically by parameterizing a resolution-agnostic kernel or by lifting to a latent space and learning integral transforms. Representative approaches include the *Fourier Neural Operator* (FNO), which performs global convolution via spectral multipliers to approximate operator kernels in Fourier space (Li et al., 2021; Kovachki et al., 2023), and *DeepONet*, which decomposes an operator into branch/trunk networks to separately encode input functions and query coordinates (Lu et al., 2021). Variants extend these ideas with multiresolution bases (Li et al., 2020b; Gupta et al., 2021; He et al., 2024; Li et al., 2026b), graph or kernelized message passing (Li et al., 2025), learned Green's functions (Li et al., 2020a), and—more recently—physics-consistent diffusion models stacked on top of operator outputs to recover sharp super-resolved fields (Li et al., 2026a). These models are *primarily data driven*: although many designs are physics-inspired, their internal representations are typically opaque to direct inspection. The learned latent bases and mixing weights are not, in general, accompanied by explicit geometric or modal parameters, which limits diagnostic insight even when not invoking any one-to-one physical interpretation.

**Transformer-based methods.** A second line of work adopts Transformers to parameterize neural operators, replacing hand-crafted kernel parameterizations with data-driven attention. Examples include *Galerkin Transformers*, which align attention with variational forms (Cao, 2021); *GNOT* (Hao et al., 2023), which leverages attention for long-range coupling on irregular meshes; *Transolver* (Wu et al., 2024), which introduces slice-based attention for efficient global mixing; and *orthogonal-attention operators* that constrain attention to behave as a kernel integral (Xiao et al., 2024). Empirically, with sufficiently large training corpora and careful scaling, attention-based operators often match or surpass traditional neural operators in expressive power and generalization to out-of-distribution forcings and grids.

However, these gains come with two recurring limitations. **(i) Limited representation-level transparency:** standard attention weights are not anchored to explicit trial/test functions with geometric meaning, making it harder to relate predictions to identifiable modes or local windows. **(ii) Frequency bias:** global self-attention tends to emphasize low-rank, global correlations (low-frequency structure), while recovering sharp, localized, or high-frequency phenomena often requires architectural add-ons or extensive data augmentation. As a result, pure Transformer operators provide limited intermediate diagnostics and may under-represent fine-scale features without additional inductive biases.

**Gaussian and particle-style representations.** Anisotropic Gaussians have a long history as a flexible, locally supported basis. *Radial basis functions* (RBFs), including Gaussians, are classical universal approximators on $\mathbb{R}^d$ (Park & Sandberg, 1991; Buhmann, 2000), and meshfree particle methods place a small number of weighted kernels at scattered locations to evaluate a continuous field. More recently, the computer-graphics community has adopted a closely related primitive: *neural radiance fields* (NeRF) encode scenes as implicit functions queried by an MLP (Mildenhall et al., 2020), while *3D Gaussian Splatting* (3DGS) represents scenes by an explicit set of anisotropic 3D Gaussians with learned positions, covariances, and weights, achieving real-time rendering with a directly inspectable particle state (Kerbl et al., 2023).

Our Gaussian particle basis shares the explicit-state spirit of 3DGS—every atom carries learned geometric parameters $(\mu, \sigma, w)$ that can be visualized as ellipses overlaid on the field—but differs in three substantive respects. (i) *Use as a trial space.* We employ the Gaussian basis as the intermediate representation of an *operator*, i.e., the trial space for a Petrov–Galerkin update, rather than as a renderer for a static scene. (ii) *Input-adaptive parameters.* The particle parameters are predicted by an encoder per input field, so the basis is adapted to the current field rather than fitted per scene by global optimization. (iii) *Modal coupling.* Cross-particle interaction is parameterized in a learned modal space via attention, rather than by alpha-compositing in image space. To our knowledge, learning a particleized Gaussian basis as the primary state of a neural operator, and coupling it through PG-style attention, is relatively under-explored.

Closer to operator learning, prior INR/neural-field approaches (Serrano et al., 2023; 2024) also produce continuous, mesh-agnostic representations, but their latent state is typically a global vector with no per-site geometric structure, so individual coordinates do not carry direct geometric meaning. Multiresolution operator families (Gupta et al., 2021; He et al., 2024; Li et al., 2026b) use *fixed* multiwavelet or multigrid bases, in contrast to our *learned* anisotropic Gaussian atoms whose locations and scales adapt to the input. We view these design axes—explicit vs. opaque latent state, fixed vs. learned basis, and image-space vs. modal-space coupling—as complementary to the existing literature rather than competing.

## B. Expressivity of the Gaussian Field and GPO

### B.1. Expressivity of the Gaussian Field

**Lemma B.1** (Density of Gaussian mixtures). *On compact $\Omega$, finite mixtures of anisotropic Gaussians are dense in $C(\Omega)$ (and dense in $L^r(\Omega)$ for $1 \le r < \infty$). Hence for any continuous scalar field $v$ and $\varepsilon > 0$, there exist $G, \{\mu_i, \sigma_i, w_i\}_{i=1}^{G}$ such that $\|v(\cdot) - \sum_{i=1}^{G} w_i \exp(-\frac{1}{2}\|(\cdot - \mu_i)/\sigma_i\|^2)\|_\infty < \varepsilon$.*

*Sketch.* Standard universal approximation results for radial basis functions/Gaussian kernels.

*Proof.* We give a constructive proof based on Gaussian mollification and Riemann sums.

**Step 1: Approximate identity via Gaussian mollifiers.** Let $\Omega \subset \mathbb{R}^d$ be compact and let $v \in C(\Omega)$. By Tietze's extension theorem there exists $\tilde{v} \in C_c(\mathbb{R}^d)$ such that $\tilde{v}|_\Omega = v$. For $\Sigma \in \mathbb{R}^{d \times d}$ symmetric positive definite, set the (unnormalized) Gaussian

$$\phi_\Sigma(x) \;=\; \exp\Big(-\tfrac{1}{2}\, x^\top \Sigma^{-1} x\Big).$$

Let $\{\Sigma_\epsilon\}_{\epsilon \downarrow 0}$ be any family with $\|\Sigma_\epsilon\| \to 0$. Since Gaussians form an approximate identity, the *normalized* mollification $\tilde{v} * \phi_{\Sigma_\epsilon} / \int_{\mathbb{R}^d} \phi_{\Sigma_\epsilon}$ converges to $\tilde{v}$ *uniformly* on compact sets as $\epsilon \downarrow 0$ (uniform continuity of $\tilde{v}$ and standard approximate-identity properties). Because the normalization constant is a positive scalar depending only on $\Sigma_\epsilon$, we can absorb it into the mixture weights later. Hence, for any $\eta > 0$ there exists $\epsilon_0$ such that for all $0 < \epsilon \le \epsilon_0$,

$$\sup_{x \in \Omega} \Big| (\tilde{v} * \phi_{\Sigma_\epsilon})(x) - \tilde{v}(x) \Big| < \tfrac{\eta}{2}. \tag{30}$$

**Step 2: Riemann-sum approximation of the convolution (finite mixture).** Fix such an $\epsilon$, write $\Sigma = \Sigma_\epsilon$, and denote the convolution

$$(\tilde{v} * \phi_\Sigma)(x) \;=\; \int_{\mathbb{R}^d} \tilde{v}(y)\, \phi_\Sigma(x - y)\, dy.$$

Since $\tilde{v}$ is compactly supported and continuous while $\phi_\Sigma$ is continuous and rapidly decaying, the integrand is continuous with compact support in $y$ uniformly in $x \in \Omega$. Hence Riemann sums approximate the integral uniformly in $x$: there exists a finite set of nodes $\{\mu_i\}_{i=1}^{G} \subset \mathbb{R}^d$ with associated positive quadrature weights $\{\Delta_i\}_{i=1}^{G}$ such that

$$\sup_{x \in \Omega} \left| (\tilde{v} * \phi_\Sigma)(x) \;-\; \sum_{i=1}^{G} \tilde{v}(\mu_i)\, \phi_\Sigma(x - \mu_i)\, \Delta_i \right| < \tfrac{\eta}{2}. \tag{31}$$

Define mixture weights $w_i := \tilde{v}(\mu_i)\, \Delta_i$ (real-valued; the lemma does not restrict their sign), and note that each term is exactly a (shared-covariance) Gaussian atom $\exp\big(-\tfrac{1}{2}\|(x - \mu_i)\|_{\Sigma^{-1}}^2\big)$, i.e.,

$$\sum_{i=1}^{G} w_i\, \phi_\Sigma(x - \mu_i) \;=\; \sum_{i=1}^{G} w_i \exp\Big(-\tfrac{1}{2}(x - \mu_i)^\top \Sigma^{-1} (x - \mu_i)\Big).$$

**Step 3: Uniform approximation on $\Omega$.** Combining (30) and (31),

$$\sup_{x \in \Omega} \left| \tilde{v}(x) - \sum_{i=1}^{G} w_i \exp\Big(-\tfrac{1}{2}(x - \mu_i)^\top \Sigma^{-1}(x - \mu_i)\Big) \right|$$
$$\le \sup_{x \in \Omega} \left| \tilde{v}(x) - (\tilde{v} * \phi_\Sigma)(x) \right| + \sup_{x \in \Omega} \left| (\tilde{v} * \phi_\Sigma)(x) - \sum_i w_i \phi_\Sigma(x - \mu_i) \right|$$
$$< \eta.$$

Restricting back to $\Omega$ (where $\tilde{v} = v$) yields

$$\Big\| v(\cdot) - \sum_{i=1}^{G} w_i \exp\big(-\tfrac{1}{2}\|(\cdot - \mu_i)\|_{\Sigma^{-1}}^2\big) \Big\|_\infty < \eta.$$

Since $\eta > 0$ was arbitrary, finite mixtures of (possibly anisotropic) Gaussians are dense in $C(\Omega)$.

**Anisotropy and vector-valued extension.** We used a common covariance $\Sigma$ for clarity; allowing mode-dependent $\Sigma_i$ only increases expressivity, so the same result holds with per-atom anisotropy. For vector-valued $v$, apply the scalar result componentwise.

**$L^r$ density.** Because $C(\Omega)$ is dense in $L^r(\Omega)$ for $1 \leq r < \infty$ on compact $\Omega$, the uniform approximation implies $L^r$ approximation, completing the proof. $\qquad\square$

## B.2. Expressivity of GPO

**Theorem B.2** (Universal approximation in modal form). *Let $\mathcal{T} : \mathcal{X} \to \mathcal{Y}$ be a continuous operator on compacta that admits a Hilbert–Schmidt (Mercer-type) kernel $K(\mathbf{x}, \mathbf{x}')$ or, more generally, a low-rank factorization $\mathcal{T} \approx \Phi(\cdot)\, \mathcal{K}\, \Phi(\cdot)^\top$ with continuous features $\Phi : \Omega \to \mathbb{R}^m$. Then, for any $\varepsilon > 0$, there exist $G$ and network parameters $\Theta$ such that $\|\mathcal{G}_\Theta - \mathcal{T}\|_{\mathcal{X} \to \mathcal{Y}} < \varepsilon$.*

*Sketch.* By Lemma B.1 and universal approximation of MLPs, windows $p(\mathbf{x}, g)$ and latent features $S(Z(\mathbf{x}))$ approximate $\Phi(\mathbf{x})$; attention realizes a trainable $\mathcal{K}$ on the $G$ modes. The scatter-and-decoder emulate the output feature map. Increasing $G$ and widths yields density in the space of continuous operators.

*Proof.* We prove the claim for operators on compact domains by reducing to a finite–rank Mercer approximation and showing that each stage of our pipeline can approximate the corresponding finite–dimensional objects arbitrarily well. Throughout, $\|\cdot\|_{\mathcal{X} \to \mathcal{Y}}$ denotes the operator norm on bounded subsets.

**Step 0: Mercer (or low–rank) truncation.** Assume $\mathcal{T}$ is continuous on bounded sets and admits either a Hilbert–Schmidt kernel $K(\mathbf{x}, \mathbf{x}')$ or, more generally, a low–rank factorization $\mathcal{T} \approx \Phi(\cdot)\, \mathcal{K}\, \Phi(\cdot)^\top$ with continuous $\Phi : \Omega \to \mathbb{R}^m$. In the Mercer case, by spectral theory,

$$K(\mathbf{x}, \mathbf{x}') = \sum_{r=1}^{\infty} \lambda_r\, \varphi_r(\mathbf{x})\, \varphi_r(\mathbf{x}'), \quad \lambda_r \geq 0, \ \{\varphi_r\} \subset C(\Omega),$$

and the partial sums define finite–rank operators $\mathcal{T}_m f(\mathbf{x}) = \sum_{r=1}^{m} \lambda_r \varphi_r(\mathbf{x}) \int \varphi_r(\mathbf{x}') f(\mathbf{x}')\, d\mathbf{x}'$ with $\|\mathcal{T} - \mathcal{T}_m\| \to 0$ as $m \to \infty$ (uniform on compacta). In the given low–rank form, select $m$ and continuous $\Phi_m : \Omega \to \mathbb{R}^m$, $\mathcal{K}_m \in \mathbb{R}^{m \times m}$ such that

$$\left\|\mathcal{T} - \mathcal{T}_m\right\| < \varepsilon/3, \qquad \mathcal{T}_m f(\mathbf{x}) = \Phi_m(\mathbf{x})\, \mathcal{K}_m \int_\Omega \Phi_m(\mathbf{x}')^\top f(\mathbf{x}')\, d\mathbf{x}'. \tag{32}$$

**Step 1: Approximating the feature maps by Gaussian basis + MLPs.** By Lemma B.1 (density of Gaussian mixtures) and universal approximation of MLPs, for any $\delta > 0$ there exist: (i) a pointwise encoder/evaluator producing $Z(\mathbf{x}) \in \mathbb{R}^G$ from Gaussian particles $(\mu, \sigma, w)$ and a small MLP $S$ such that the *trial features* $\Psi(\mathbf{x}) \in \mathbb{R}^D$, defined by $\Psi(\mathbf{x}) = S(Z(\mathbf{x}))$, satisfy

$$\sup_{\mathbf{x} \in \Omega} \left\|\Psi(\mathbf{x}) - \Phi_m(\mathbf{x})\right\|_2 < \delta; \tag{33}$$

(ii) head–wise *Gaussian modal windows* $p(\mathbf{x}, g) \geq 0$ with $\sum_{g=1}^{G} p(\mathbf{x}, g) = 1$, implemented by linear maps on $[Z(\mathbf{x}), (\mu, \sigma, w)(\mathbf{x})]$ and a softmax, such that the *test* functionals

$$\mathcal{M}_g(f) = \int_\Omega p(\mathbf{x}, g)\, f(\mathbf{x})\, d\mathbf{x}$$

approximate the $m$ target coordinates $\int \Phi_m(\mathbf{x})^\top f(\mathbf{x})\, d\mathbf{x}$ after a fixed linear readout. Concretely, there exists $W \in \mathbb{R}^{m \times G}$ with $\| W\, [\mathcal{M}_g(\cdot)]_{g=1}^{G} - \int \Phi_m(\cdot)^\top(\cdot) \| < C_1 \delta$. (One can view $W p(\mathbf{x}, \cdot)$ as a learned quadrature/test family for the $m$ coordinates.)

**Step 2: Discrete PG measurement and quadrature error.** Given a discretization $\{\mathbf{x}_j\}_{j=1}^{N}$ with empirical measure converging to the sampling measure on $\Omega$, the $N \to G$ aggregation used in Sec. 3.2 forms tokens

$$t_g = \frac{\sum_{j=1}^{N} p(\mathbf{x}_j, g)\, \Psi(\mathbf{x}_j)}{\sum_{j=1}^{N} p(\mathbf{x}_j, g)} \quad \in \mathbb{R}^D.$$

By uniform continuity of $\Psi$ and $p(\cdot, g)$ on compact $\Omega$, Riemann (or Monte Carlo) sums converge to the integrals. Hence there exists $N_0$ so that for all $N \geq N_0$,

$$\left\| \left[ t_g \right]_{g=1}^{G} - \left[ \frac{\int p(\mathbf{x}, g)\, \Psi(\mathbf{x})\, d\mathbf{x}}{\int p(\mathbf{x}, g)\, d\mathbf{x}} \right]_{g=1}^{G} \right\| < C_2 \delta. \tag{34}$$

Post-multiplying by $W$ and using (33) shows that the vector of $m$ measured coordinates is within $C_3 \delta$ of $\int \Phi_m(\mathbf{x})^\top f(\mathbf{x})\, d\mathbf{x}$ for any $f$ in a bounded set.

**Step 3: Implementing the modal coupling by attention + linear maps.** We next show that the $G \times G$ *modal attention* stage can realize the finite linear map $\mathcal{K}_m$ (up to basis changes) to arbitrary precision. Using the head projections $W_z$ and $W_{\mathrm{out}}$, the attention block computes

$$\widetilde{T} = \alpha \left( T W_V \right), \qquad Y = \left( \widetilde{T} \right) W_{\mathrm{out}},$$

where $T \in \mathbb{R}^{G \times D}$ stacks the tokens $t_g$, $\alpha$ is the softmax attention matrix, and $W_V, W_{\mathrm{out}}$ are learned linear maps. Since softmax can approximate a Kronecker–delta (by sending on–diagonal logits to $+\infty$ and off–diagonal to $-\infty$), we can set $\alpha \approx I_G$ arbitrarily closely. Then $Y \approx T(W_V W_{\mathrm{out}})$. Because $W_V, W_{\mathrm{out}}$ are unconstrained, their product can approximate any target matrix $M \in \mathbb{R}^{D \times m}$ to arbitrary precision. Choosing $M$ to implement the composition $W \mathcal{K}_m$ (after the measurement map from Step 2), we obtain a block that emulates $v \mapsto \mathcal{K}_m v$ in the $m$–dimensional modal coordinates. (If desired, one may keep $\alpha$ nontrivial and absorb its effect into the surrounding linear maps; the argument is unchanged.)

**Step 4: Scatter and pointwise decoding.** The $G \to N$ scatter re-distributes the mixed modal features back to locations via the same windows $p(\mathbf{x}, g)$, followed by a pointwise decoder MLP $f_\phi^{\mathrm{dec}} : \mathbb{R}^G \to \mathbb{R}^{c_{\mathrm{out}}}$. Since MLPs are universal approximators on compacta, the composition can approximate the desired output feature map $\mathbf{x} \mapsto \Phi_m(\mathbf{x})$ (or its linear image) uniformly, matching the form in (32).

**Step 5: Error aggregation.** Let $\varepsilon_m = \|\mathcal{T} - \mathcal{T}_m\| < \varepsilon/3$ be the truncation error. Pick $\delta > 0$ sufficiently small and $N$ sufficiently large so that: (i) the feature/window approximations introduce at most $C\delta$ error in the measured coordinates (Steps 1–2), (ii) the attention+linear block approximates the modal coupling $\mathcal{K}_m$ within $C\delta$ uniformly on bounded sets (Step 3), and (iii) the scatter+decoder approximates the output features within $C\delta$ uniformly (Step 4). By stability (continuity) of all stages,

$$\left\| \mathcal{G}_\Theta - \mathcal{T} \right\| \leq \underbrace{\left\| \mathcal{G}_\Theta - \mathcal{T}_m \right\|}_{\leq C\delta} + \underbrace{\left\| \mathcal{T}_m - \mathcal{T} \right\|}_{\varepsilon_m} < C\delta + \varepsilon/3.$$

Choosing $\delta$ so that $C\delta < 2\varepsilon/3$ yields $\|\mathcal{G}_\Theta - \mathcal{T}\| < \varepsilon$.

Combining the steps completes the proof. $\qquad\square$

## C. Implementation Details

### C.1. Baseline implementations

Baseline models (Geo-FNO, LSM, Galerkin Transformer, GNOT, ONO, Transolver) are adapted from the *Neural-Solver-Library* (Wu et al., 2024) reference implementation at `https://github.com/thuml/Neural-Solver-Library`. Other models are adapted from their official repositories. Unless otherwise noted, we keep an identical training schedule across baselines: AdamW optimizer, initial learning rate $10^{-3}$ with a `StepLR` scheduler (step size and decay factor as in the library's default per dataset), up to 500 epochs with validation early stopping, the same data normalization/inverse-normalization protocol, and matched rollout/evaluation settings.

### C.2. GPO configurations

The dataset-specific configurations of GPO are summarized in Tab. 2. We provide the source code of GPO in the Supplementary Material.

*Table 2.* **Model configurations of GPO.**

| BENCHMARKS | MODEL CONFIGURATIONS | | | |
|---|---|---|---|---|
| | HIDDEN_DIM | NUM_LAYERS | NUM_HEADS | NUM_GAUSSIANS |
| NS2D | 128 | 8 | 8 | 32 |
| NS3D | 64 | 8 | 4 | 16 |
| ERA5-TEMP | 64 | 4 | 4 | 16 |
| ERA5-WIND U | 64 | 4 | 4 | 16 |
| CARRA | 64 | 4 | 4 | 16 |
| AIRFOIL | 64 | 4 | 4 | 16 |
| TURBULENT | 128 | 8 | 8 | 32 |
| PLANETSWE | 64 | 8 | 4 | 16 |

## D. Model Analysis

### D.1. Ablation Study

Tab. 3 reports the one-step prediction $L_2$ error on NS2D under controlled variants (parameter counts adjusted to be comparable). **(i) Role of the Gaussian Field.** Replacing the Gaussian Field with a plain MLP encoder/decoder (`w/o Gaussian Field`) degrades accuracy to $7.44\times10^{-2}$, and removing the PG operator while keeping the Gaussian Field (`w/o PG Operator`) gives $8.57\times10^{-2}$. Even a single Gaussian per site (`num_gaussian=1`) already improves to $6.28\times10^{-2}$, suggesting the particleized Gaussian evaluation provides a useful inductive bias beyond a black-box MLP. **(ii) Combining PG operator and Gaussian Field.** Combining the Gaussian basis with PG Gaussian Attention yields the full GPO (NS2D baseline: $3.90\times10^{-2}$), indicating that the PG measurement→modal coupling→scatter is complementary to the local particle representation. **(iii) Effect of the number of Gaussians.** Increasing `num_gaussian` consistently reduces error (from $4.21\times10^{-2}$ at $G$=16 to $3.84\times10^{-2}$ at $G$=64), with diminishing returns; we adopt $G$=16/32 for an accuracy–efficiency trade-off (see Sec. D.5).

*Table 3.* **Ablation study** on NS2D one-step prediction. Relative $L_2$ error of GPO variants.

| MODEL CONFIGURATION | $L_2$ ERROR |
|---|---|
| W/O PG OPERATOR | 8.57E-02 |
| W/O GAUSSIAN FIELD | 7.44E-02 |
| NUM_GAUSSIAN = 1 | 6.28E-02 |
| NUM_GAUSSIAN = 16 | 4.21E-02 |
| NUM_GAUSSIAN = 64 | 3.84E-02 |
| GPO (BASELINE) | 3.90E-02 |

### D.2. Kernel Ablation

Tab. 4 replaces the Gaussian basis evaluator with alternative irregular-point-compatible kernels (Laplacian, multiquadric RBF, inverse-quadric RBF), keeping the encoder, PG operator, and decoder unchanged. The Gaussian basis attains the best overall $L_2$ on NS2D; alternative kernels are competitive but consistently behind. We therefore retain anisotropic axis-aligned Gaussians as the default.

*Table 4.* **Kernel ablation** under the same irregular-point operator pipeline (NS2D, one-step prediction). We replace the Gaussian basis evaluator with alternative kernels while keeping the encoder, operator, and decoder unchanged. The Gaussian basis yields the most consistent overall performance.

| BASIS / KERNEL | $L_2$ ERROR |
|---|---|
| LAPLACIAN | 5.18E-02 |
| MULTIQUADRIC RBF | 5.27E-02 |
| INVERSE QUADRIC RBF | 4.86E-02 |
| **GAUSSIAN (OURS)** | **3.90E-02** |

### D.3. Center-Regularizer Sensitivity

Tab. 5 sweeps the strength of the center regularizer $\mathcal{L}_\mu$. Removing $\mathcal{L}_\mu$ gives the worst center alignment and also degrades the scale regularization $\mathcal{L}_\sigma$ that the encoder otherwise learns implicitly. A relaxed $\mathcal{L}_\mu$ ($0.1\times$) can marginally improve $L_2$ while losing some geometric regularity; a strong $\mathcal{L}_\mu$ ($5\times$) gives the best $\mathcal{L}_\mu$ value but mildly hurts $L_2$. The default $1\times$ setting offers the best overall trade-off and is what we use throughout the paper.

*Table 5.* **Ablation on the center regularizer** $\mathcal{L}_\mu$ (NS2D, one-step prediction). Removing $\mathcal{L}_\mu$ leads to the worst center alignment and degrades scale regularity; relaxing $\mathcal{L}_\mu$ slightly improves $L_2$ in some cases, while the default setting gives the best overall trade-off.

| $\mathcal{L}_\mu$ CONFIGURATION | $L_2$ ERROR | $\mathcal{L}_\mu$ | $\mathcal{L}_\sigma$ |
|---|---|---|---|
| W/O $\mathcal{L}_\mu$ | 6.32E-02 | 1.87E-02 | 7.43E-03 |
| RELAXED $\mathcal{L}_\mu$ ($0.1\times$) | 3.84E-02 | 8.12E-03 | 3.46E-03 |
| STRONG $\mathcal{L}_\mu$ ($5.0\times$) | 4.08E-02 | **2.76E-03** | 3.28E-03 |
| **DEFAULT $\mathcal{L}_\mu$ ($1.0\times$)** | **3.90E-02** | 4.35E-03 | **2.91E-03** |

### D.4. Capacity-Fair Comparison with FNO

To rule out capacity as a confound, we re-train FNO with two stronger configurations on the Turbulent task: a wider variant (channel width $128 \to 256$, FC dim $256 \to 512$) and a more-modes variant (mode budget per direction increased from $[15, 12, 9, 9, 9]$ to $[21, 18, 15, 15, 15]$). Both variants close some of the gap to GPO, with the more-modes variant being more effective, but neither reaches GPO's error at a similar or larger parameter budget (Tab. 6). This indicates that GPO's advantage on multiscale tasks is not solely driven by parameter count.

*Table 6.* **Capacity-fair comparison against stronger FNO baselines** on the Turbulent task (one-step prediction). Scaling width yields only marginal gains, while increasing the number of retained Fourier modes helps more but still does not close the gap to GPO, indicating that the improvement is not solely attributable to parameter count.

| MODEL | PARAMS | $L_2$ ERROR | MODES ($m_1, m_2$) | WIDTH | FC DIM | LAYERS |
|---|---|---|---|---|---|---|
| FNO (ORIGINAL) | 648,625 | 4.82E-01 | $[15, 12, 9, 9, 9]$ | 128 | 256 | 16-24-24-32-32 |
| FNO (WIDER) | 1,318,241 | 4.78E-01 | $[15, 12, 9, 9, 9]$ | 256 | 512 | 24-32-32-48-48 |
| FNO (MORE MODES) | 1,537,969 | 4.63E-01 | $[21, 18, 15, 15, 15]$ | 128 | 256 | 16-24-24-32-32 |
| **GPO (OURS)** | 1,598,257 | **3.95E-01** | / | / | / | / |

## D.5. Computational Complexity

Empirical measurements (Tab. 7, $64\times64\times3$, batch 16) corroborate the analysis: GPO attains a low memory footprint (2,313 MiB) and competitive time (44.66 s/epoch train; 1.67 s/epoch inference) with a modest parameter count (6.10 MB), outperforming attention baselines in training speed (Galerkin/Transolver/ONO/GNOT) and GPU memory, while remaining close to spectral baselines at inference. Although FNO is fastest on this small grid, GPO's cost grows near-linearly with $N$ and remains stable when moving to higher resolutions or 3D, where spatial attention becomes prohibitive and FFT memory/I/O costs rise.

By aggregating *locally* ($N \leftrightarrow G$) and coupling *globally* only in modal space ($G\times G$), GPO delivers resolution-agnostic efficiency: linear scaling in $N$, controllable quadratic dependence on $G$, and favorable memory/time trade-offs across 2D/3D and irregular domains.

*Table 7.* **Computational efficiency comparison across models** (measured with input size $64\times64\times3$, batch size 16).

| MODEL | PARAM COUNT | PARAM (MB) | GPU MEM (MIB) | TRAIN (S/EPOCH) | INFERENCE (S/EPOCH) |
|---|---|---|---|---|---|
| FNO | 640,305 | 4.84 | 949 | 28.27 | 0.5 |
| LSM | 19,187,457 | 73.23 | 2,875 | 48.42 | 1.73 |
| GALERKIN TRANSFORMER | 1,096,321 | 4.18 | 4,301 | 65.29 | 2.96 |
| GNOT | 2,485,901 | 9.48 | 8,643 | 139.42 | 6.09 |
| TRANSOLVER | 3,069,889 | 11.71 | 4,917 | 97.03 | 4.10 |
| ONO | 1,596,673 | 6.09 | 6,163 | 94.80 | 4.27 |
| **GPO (OURS)** | 1,598,257 | 6.10 | 2,313 | 44.66 | 1.67 |

# E. Additional Visualizations

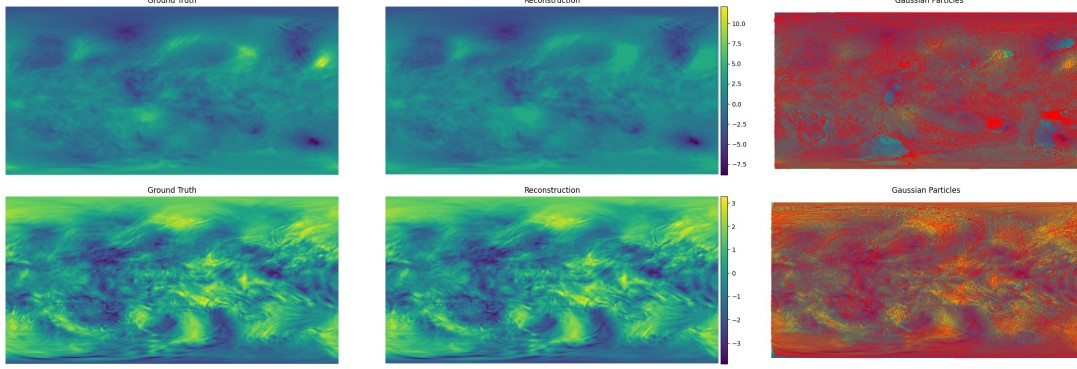

*Figure 7.* **Reconstruction setting** on ERA5: representation-level visualization on an **in-distribution sample (top)** and an **out-of-distribution sample (bottom)**. Left to right: ground-truth field, reconstruction from the Gaussian basis, and learned Gaussian particles overlaid (ellipses: center $\mu$, axes $\propto \sigma$, color/size $\propto w$).

