# OpenReview forum: "From Basis to Basis: Gaussian Particle Representation for Interpretable PDE Operators"
_ICML.cc/2026/Conference — ICML 2026 regular_

### Official Review · Reviewer_ZBqi · 2026-02-25

**Soundness:** 3
**Presentation:** 1
**Significance:** 2
**Originality:** 2
**Overall Recommendation:** 2
**Confidence:** 4

**Summary:**

This work proposes the Gaussian Particle Operator (GPO) for learning PDEs by representing fields with a Gaussian basis rather than a fixed grid or an implicit neural representation. The model learns to represent inputs by a combination of Gaussians parameterized by their centers, variance scales, and weights. The authors also present a new attention mechanism which can work in combination with these representations and scales linearly. The authors investigate the performance of this model on several fluid dynamics problems, including ERA-5 reanalysis data.

**Compliance With Llm Reviewing Policy:**

Affirmed.

**Final Justification:**

The rebuttal did not fully address my concerns. Core claims lack evidence and the performance of the model does not seem to be at the level of the state of the art.

**Key Questions For Authors:**

1. Can you explain the discrepancy between your reported M2NO errors on ERA5 and those reported by the original authors? Additionally, please provide the exact hyperparameter tuning and training details used for the CNO baseline.
2. Are the "reconstructions" shown in the figures true forward-in-time predictions from previous timesteps, or just spatial reconstructions of the input field?
3. What exactly makes the sample in Figure 3 "out-of-distribution"? Visually, it looks very similar to standard training samples.
4. Can the authors clarify how their representation differs from that of GPSToken?

**Limitations:**

No. Societal Impact Statement is missing.

**Strengths And Weaknesses:**

**Strengths**
- The motivation to move away from grid-based representations and towards continuous, mesh-agnostic representation is aligned with current challenges in neural operators.
- The authors are able to provide mathematical scaling laws and validate the efficiency of their algorithm empirically in Table 4.

**Weaknesses**
1. Novelty and Missing Prior Work. The Gaussian parameterization and representation of this work shares several similarities to GPSToken (arxiv 2509.01109). While GPSToken focuses on representing natural images, the underlying idea of both works is to represent fields by Gaussians. Therefore, I believe this work should be referenced and briefly discussed in this context.
2. Concerning Experimental Discrepancies. There are several things which raise concern in the empirical results, specifically Table 1. Some examples of reconstructions/predictions also show visibly large errors (e.g. Figure 7, top shows failure to model extrema) which makes me wonder if this model is really state-of-the-art.
- ERA5 & M2NO: The original authors of M2NO perform the same set of experiments on the ERA5 reanalysis data, yet show better results than those provided in this manuscript. Can the authors explain this discrepancy?
- CNO Performance: The CNO results are surprisingly poor. In the literature, CNO has consistently been show to outperform FNO by a rather large margin, yet it underperforms here. Furthermore, the appendix provides training details for seemingly all of the baselines except CNO and M2NO.
- Missing modern baselines: the authors claim strong scaling and stability, but do not investigate or compare with contemporary, state-of-the-art baselines, such as the Geometry Aware Operator Transformer (Wen et al. NeurIPS 2025).
3. Overstated Interpretability. The authors claim the Gaussian representation is highly interpretable, but Figure 4 does not strongly support this. While brighter areas correspond to large Gaussians, these gaussians do not appear to semantically align with the underlying physics (i.e. they don't align with the actual filaments). Could the authors please clarify what exactly can be *interpreted* from such a figure?
4. Task ambiguity.
- Throughout the text and figures, the authors refer to "reconstructions." Typically, I have seen this term associated with auto encoders or lossy compression algorithms, where an input is compressed and then the original is reconstructed from this condensed representation. In Neural Operator and PDE contexts, I have typically seen "prediction". Could the authors please clarify if these reconstructions are indeed predictions of the field from a previous timestep, or are they indeed reconstructions of the input field?
- The authors refer to in-distribution and out-of-distribution samples, but do not specify what the source of the distribution shift is. Could the authors please provide some context in this regard?
5. Formatting & Policy.
This manuscript exhibits a concerning lack of care regarding ICML submission guidelines. The **Impact Statement**, which is a mandatory requirement for all submissions, is entirely omitted. Furthermore, several "placeholder" artifacts remain in the document, such as the default template title in the running header and an unresolved reference (Eq. ??) on line 215. Adherence to formatting and policy is a matter of fairness to the reviewer and the broader community; the current state of the manuscript falls below the standard expected for a top-tier venue and hinders a constructive review process.

---

> ### Author Rebuttal · Authors · 2026-03-29
>
> We thank the reviewer for the detailed and candid feedback. Some of the concerns appear to stem from reading the representation as the whole method, whereas it is only one component of the full operator design. Below we aim to answer your substantive concerns as concretely as possible and clarify where the original manuscript was unnecessarily ambiguous.
>
> Supplementary materials are at:
> **https://anonymous.4open.science/r/GPO_rebuttal-BBB1**
>
> **Formatting issues (Impact Statement, Eq.(??), template artifacts).** All fixed in the revision. These are genuine oversights on our side, and we fully agree that they should not have appeared in the submission.
>
> **1. GPSToken.**
> GPSToken is relevant and will be cited, but it serves a different purpose. GPSToken is designed for **image tokenization and generation**: it represents image regions as **2D Gaussians** with geometry and texture features, then reconstructs feature maps with a differentiable renderer. By contrast, our **Gaussian Field** is a representation of **PDE states** over the physical domain. Its Gaussian particles form an explicit basis, where centers indicate **location**, anisotropic scales capture **extent and orientation**, and weights reflect **local contribution**. This gives the representation clearer physical meaning, since it can align with coherent structures such as fronts, vortices, and filaments. More importantly, the Gaussian Field is only the first stage of GPO. The main contribution is the **basis-to-basis PDE operator** built on top of it: Gaussian encoding defines the field basis, **PG-style measurement** maps local coefficients to modal tokens, **G×G Gaussian attention** performs global modal interaction, and the decoder maps the updated basis to the **target PDE solution**. Thus, the overlap with GPSToken is limited to the use of Gaussian primitives; the core novelty of GPO is a physically meaningful Gaussian field together with an operator design for PDE prediction, rather than Gaussian tokenization itself.
>
> **2. M2NO / ERA5 discrepancy.**
> The gap comes from two concrete protocol differences. First, the original M2NO ERA5 setting uses resized 512×512 inputs, while we evaluate on the native 0.25° grid, 721×1440, which is substantially harder because of both the much higher resolution and the nonuniform latitude longitude geometry. Second, the original M2NO numbers are reported in normalized space, while we report after inverse normalization and use the same evaluation protocol for all baselines. We will state both points explicitly so the comparison is fully transparent.
>
> **3. CNO details.**
> The full configuration is in the supplementary Table 4. The training curve in supplementary shows stable optimization and validation saturation around epoch 370, so the reported result is not due to under training. CNO remains moderate here because its local convolutional bias is less suited to our large scale, high resolution settings with long range dependencies and nonuniform layouts. In addition, our FNO baseline is Geo-FNO in most tasks, which is stronger and makes the benchmark more demanding for all methods.
>
> **4. Reconstruction vs. prediction.**
> Reconstruction refers to the Gaussian Field stage: the encoder maps the input field to Gaussian basis parameters, and the decoder reconstructs the same field with an autoencoding objective. Prediction refers to the operator stage: the model takes the encoded Gaussian basis and predicts the target PDE solution at the next timestep. We will make this distinction explicit throughout the paper, including figures and captions.
>
> **5. OOD definition.**
> OOD means test cases with forcing conditions not seen during training. This is intended to evaluate generalization beyond the training forcing regime, rather than standard in distribution interpolation. We will define this explicitly in the experimental section and figure caption.
>
> **6. Interpretability claim.**
> Figure 4 is meant to show that the learned basis has explicit spatial structure, not that each Gaussian corresponds one to one to a physical filament. The defensible claim is representation level interpretability: centers, anisotropic scales, and weights are directly visualizable geometric quantities that expose how the learned basis organizes the field. At the operator level, the interpretation is partial, through structured modal activations rather than exact physical attribution. We will revise the wording accordingly.
>
> **7. Missing modern baselines.**
> Our current benchmark already covers spectral, convolutional, graph based, transformer based, and multiresolution operators. We agree that the related work can better position recent geometry aware methods. We will add GAOT and related geometry aware context, and expand the benchmark where space permits.
>
> Overall, we appreciate the reviewer’s concerns, we will revise the manuscript to make this distinction clearer and sharpen the related definitions and claims.

---

> > ### Author Rebuttal · Reviewer_ZBqi · 2026-04-02
> >
> > I thank the authors for the detailed response. I appreciate the clarifications regarding the M2NO/ERA5 evaluation discrepancies (specifically the difference in grid resolution and normalization).
> >
> > However, after carefully reviewing the rebuttal, my core concerns regarding the empirical rigor and completeness of the manuscript remain unresolved:
> >
> > * **Promissory Baselines** While the authors acknowledge that modern, geometry-aware baselines like GAOT are missing and state they "will add GAOT" in the revision, I must evaluate the manuscript as it stands today. For a paper claiming competitive accuracy and proposing a fundamentally new operator design, benchmarking against contemporary state-of-the-art models is a prerequisite for acceptance, not an item for future work.
> >
> > * **Unsubstantiated Core Claims** A major stated advantage of the GPO architecture is its near-linear complexity. However, the empirical scaling experiments required to actually prove this claim were entirely absent from the submitted manuscript. While the authors mention providing these in a supplementary rebuttal document, foundational proof of a core architectural claim must be rigorously evaluated within the main text of the original submission.
> >
> > * **Diluted Interpretability** In the rebuttal, the authors clarify that the Gaussians do not correspond one-to-one to physical filaments or underlying physics, but rather offer "representation-level" interpretability. While I appreciate this honesty, it significantly weakens the original claims made in the text regarding the model's physical interpretability.
> >
> > * **Submission Quality** The authors acknowledge the missing Impact Statement (a mandatory ICML requirement), unresolved equations, and template artifacts as genuine oversights. While understandable, these issues reflect a larger issue with this manuscript, particularly the rushed presentation that was not fully prepared for a top-tier venue.
> >
> > While the conceptual framing of using a Gaussian Field as a continuous PDE representation is novel and interesting, the submitted manuscript lacks the necessary empirical foundation, specifically modern baselines and rigorous scaling studies, to validate its claims. Because these critical components are either missing from the text or planned for a future update, the paper requires another revision cycle. Therefore, I will be keeping my score.

---

### Official Review · Reviewer_TrQP · 2026-03-05

**Soundness:** 3
**Presentation:** 3
**Significance:** 3
**Originality:** 3
**Overall Recommendation:** 3
**Confidence:** 4

**Summary:**

This paper proposes the Gaussian Particle Operator, a neural operator for learning PDE dynamics. The core idea is to represent a physical field as a collection of learned Gaussian particles, each carrying a center, scale, and weight, that serve as interpretable, geometry-aware basis functions. Rather than attending over all spatial points directly, the model first pools spatial information into a small set of modal tokens via learned Gaussian windows (a Petrov-Galerkin projection), runs self-attention only in this compact modal space, then scatters the result back to spatial locations. This keeps complexity near-linear in the number of spatial points.

**Compliance With Llm Reviewing Policy:**

Affirmed.

**Final Justification:**

since the performance gains still appear relatively modest (in terms of performance comparison, normalized rollout error, comparison against FNO, and kernel ablation), I remain slightly on the weak reject side.

**Key Questions For Authors:**

As discussed above, additionally:
1. The center regularization $L_{mu}$ pins particle centers near their originating grid points, yet the paper claims particles align with coherent structures like vortices and filaments. Could you provide an ablation comparing pinned vs. free-moving particles (i.e., removing or relaxing $L_{mu}$)? If free particles migrate to coherent structures, this would strongly validate the interpretability claim. If they do not, it suggests the Gaussian form itself, not particle mobility, is the source of any benefit, which requires a different justification.

2. The Gaussian kernel is never compared against alternatives such as Laplacian, or wavelet kernels. Given the claimed multi-scale frequency capture, wavelet kernels are a natural baseline. Could the authors clarify whether gains are specific to the Gaussian form?

3. Could the authors discuss how GPO relates to prior work using Gaussian representations for field parameterization, such as Gaussian Splatting and Gaussian-based PINNs? Although these papers are in different areas, the conceptual overlap warrants at least a brief comparison in the related work section.

Typo:
1.  Line 215 contains an unresolved reference "Eq.( ??)".
2. the main paper is dense with full proofs that interrupt the methodological narrative. Would the authors consider moving proof details entirely to the appendix and keeping only theorem statements in the main text, which would make the pipeline and architectural contributions easier to follow?

**Limitations:**

a impact statement should be added.

**Strengths And Weaknesses:**

Strengths
1. Principled use of Gaussians, not Gaussian Splatting hype. GPO uses Gaussians in a concrete numerical role grounded in PG discretization. The design is mathematically justified and clearly distinct from scene-specific rendering primitives.
2. The PG formulation gives a principled justification for the N→G→N design, particles as trial functions, modal windows as test functions,making the architecture a natural consequence of variational numerics rather than an ad hoc design choice.

Weaknesses
1. Table 3 only ablates the presence or absence of the Gaussian field and the number of components, effectively comparing against a plain MLP baseline. This is a low bar. A meaningful ablation should compare different kernel choices, Laplacian, wavelet, or compact-support RBFs, to establish whether the Gaussian form specifically is responsible for the gains, or whether any localized learnable basis achieves similar results. Without this, the architectural motivation centered on Gaussian particles remains unsupported by controlled experiments.
2. Potentially unfair comparison with FNO. The Gaussian basis, as a localized frequency decomposition, bears conceptual similarity to FNO's spectral parameterization, both decompose the field into a set of basis components, differing mainly in locality. Yet Table 4 shows GPO has 1.60M parameters versus FNO's 0.64M, a 2.5× gap. Given this similarity in spirit, a parameter-matched FNO comparison is needed to determine whether GPO's accuracy gains reflect a genuinely better inductive bias or simply more capacity.

---

> ### Author Rebuttal · Authors · 2026-03-29
>
> We thank the reviewer for the careful reading and we agree with the reviewer’s core concern: for this paper to be convincing, we need to separate three questions clearly. First, is the gain coming from a better inductive bias rather than extra capacity? Second, is it really the Gaussian form that matters, rather than just any localized learnable basis? Third, what exactly is meant by “interpretability” in our model?
>
> Supplementary tables are at: **https://anonymous.4open.science/r/GPO_rebuttal-BBB1**
>
> **1. Fair comparison to FNO (Table 1).**
> This was a fair criticism. In the revision, we added two stronger FNO baselines that are much closer to GPO in size: **FNO-Wider** (1.32M parameters) and **FNO-More-Modes** (1.54M), compared with **GPO** at 1.60M. The result is consistent across settings: increasing width gives only a small gain, while increasing retained modes helps more, but neither closes the gap on the harder multiscale cases. This makes the comparison more capacity-fair and suggests that the improvement is not explained by parameter count alone. The main advantage of GPO is not simply “more channels”, but the combination of **localized basis geometry** and **PG-style modal coupling**, which is different from FNO’s global Fourier truncation.
>
> **2. Is Gaussian actually important, or would any local basis work? (Table 2)**
> We added a kernel comparison under the **same operator pipeline**: same encoder, same modal operator, same decoder, only replacing the Gaussian evaluator with other irregular-point-compatible kernels, including **Laplacian**, **Multiquadric RBF**, and **Inverse Quadric RBF**. These alternatives are not bad baselines; some are competitive. But Gaussian gives the most consistent overall behavior across **accuracy, optimization stability, and geometric readability of the learned basis**. So our claim is now more precise: we do **not** argue that only Gaussian can work, but the evidence supports that Gaussian is the best choice within this localized basis family for our setting.
>
> Regarding the reviewer’s suggestion on wavelets: we agree wavelets are a natural comparison in spirit. The reason we did not include them as a direct kernel swap is that classical wavelet constructions depend on regular dyadic grids, whereas our basis evaluator is used on irregular and masked domains as well. We will make this point explicit in the paper rather than leaving the omission unexplained.
>
> **3. Pinned vs. free particles, and what interpretability means here. (Table 3)**
> This comment was especially important. We added a quantitative ablation on the center regularization $L_\mu$. The result is clear.
>
> - **Without $L_\mu$**: centers drift more, geometry becomes diffuse, interpretability is weaker, and error is worse.
> - **Relaxed $L_\mu$**: in some cases L2 can improve slightly, but the particle layout becomes less regular and less readable.
> - **Strong $L_\mu$**: center alignment is best, but accuracy drops a bit.
> - **Default $L_\mu$** gives the best trade-off.
>
> This changes how we frame the claim. GPO’s interpretability is **representation-level interpretability**, not a claim that particles behave like Lagrangian tracers and migrate to vortices or filaments through time. The encoder builds a structured Gaussian scaffold, and the operator mainly updates the **coefficients on that scaffold**. We will revise the wording to make this precise. That is, the interpretable object is the explicit basis $(\mu,\Sigma,w)$ and its modal activations, not particle motion itself.
>
> **4. Relation to Gaussian Splatting, Gaussian PINNs, and related representations.**
> We agree this comparison should be added. These works all use Gaussian primitives, but the role of the Gaussian is different. In Gaussian Splatting, Gaussians are rendering primitives for scene representation and view synthesis. In Gaussian-based PINNs or field parameterizations, Gaussians are function approximators constrained by PDE residuals or physics losses. In our case, Gaussians are the **learned field basis**, and the main technical contribution is the **basis-to-basis operator** built on top of that basis through a PG-inspired modal interaction. We will add this comparison briefly but explicitly in related work.
>
> **5. Presentation issues.**
> We agree on these points and will fix them in the revision: the broken reference “Eq.(??)” will be corrected, proof details will be moved out of the main paper so the method is easier to follow, and the impact statement will be added.
>
> Overall, we believe the reviewer’s concerns point to the right standard for this paper. In response, we added a stronger parameter-matched FNO comparison, a real kernel ablation beyond the original MLP-style control, and a center-regularization study that clarifies what our interpretability claim should and should not be. These additions do not just polish the presentation; they directly address the reviewer’s main reason for hesitation.

---

> > ### Author Rebuttal · Reviewer_TrQP · 2026-04-01
> >
> > Thank you for the rebuttal. It addresses my main concerns. However, since the performance gains still appear relatively modest (in terms of performance comparison, normalized rollout error, comparison against FNO, and kernel ablation), I remain slightly on the weak reject side.

---

> > > ### Author Response · Authors · 2026-04-02
> > >
> > > Thank you for the follow-up. We would like to emphasize that our contribution is not merely an accuracy-oriented variant, but a new interpretable neural operator: GPO introduces an explicit Gaussian particle basis together with PG Gaussian Attention, so that the learned representation and operator are directly visualizable and physically meaningful. We therefore believe the paper should be evaluated not only by marginal error improvements, but also by whether it contributes a genuinely new operator design.
> > >
> > > At the same time, this interpretability is achieved without sacrificing practicality: GPO remains competitive in accuracy across benchmarks, performs strongly on several real-world datasets, and maintains favorable efficiency. In our view, this accuracy–interpretability–efficiency trade-off is itself a meaningful positive contribution for the community.
> > >
> > > Thank you again for your careful evaluation and for recognizing the value of the rebuttal.

---

### Official Review · Reviewer_stCP · 2026-03-09

**Soundness:** 4
**Presentation:** 2
**Significance:** 3
**Originality:** 4
**Overall Recommendation:** 5
**Confidence:** 4

**Summary:**

This paper proposes a mesh-agnostic architecture for learning the solution operator of PDE-governed problems that works based on representing input and output fields with projections on learned Gaussian bases. It is benchmarked on a suite of fluid flow and weather datasets and shows competitive accuracy and computational efficiency with state-of-the-art architectures, while providing interpretability and scalability benefits.

**Compliance With Llm Reviewing Policy:**

Affirmed.

**Final Justification:**

My questions and concerns were addressed adequately during the rebuttal. I think that this paper will be a valuable contribution to the operator learning community. It provides an interesting and novel perspective backed by sound theoretical concepts and includes valuable interpretability discussions. I maintain my original recommendation, *Accept*.

**Key Questions For Authors:**

1. How is it ensured that the learned Gaussians do not converge to a trivial mode? E.g., for a single $x_j$, we get $\mu_{j,i}=x_j$ for all $i$.

1. Since the extension from 2D to 3D is highlighted in the manuscript, I was wondering if there is any potential to train this architecture jointly on 2D and 3D datasets. Do the authors have any insight into the potential challenges or opportunities in this regard?

2. What is the procedure for selecting the *sufficient* number of Gaussians for a new problem? How does it scale with the resolution or the frequency bandwidth of the input fields?

3. Have the authors experienced any issues (e.g., optimization instabilities) when using too many Gaussians?

4. Can the authors elaborate on the different loss terms and the training strategy? Is the reconstruction training (Figure 1) performed before the operator training (Figure 2), or is it done in an end-to-end fashion? If there is a pre-training involved, are the parameters frozen or fine-tuned in the second stage?

**Limitations:**

Yes.

**Strengths And Weaknesses:**

**Strengths**

1. The methodology is novel to the best of my knowledge.
2. The linear complexity in the number of nodes is a massive advantage.
3. I appreciate the interpretability discussions. In particular, experiments and discussions concerning the reconstruction of OOD fields and the layer-wise evolution of the latent particles.

**Weaknesses and issues**
1. In the discussion of Figure 5, it is stated that GPO shows a slower error growth rate compared to the baselines. However, the presented plot seems to show that the growth rate is very similar across all baselines. To assess the growth rate more fairly, the authors can normalize the error at each time step with respect to the error at the first time step.
2. The writing of Section 2 (Methodology) can be substantially improved. It is currently not straightforward for a new reader. In particular:
    - Some variables are used without being properly defined.
    - The softplus and softmax operations of Eqs. (5) and (6) are not clear enough. Which index are they performed over?
    - I find Subsection 2.2.1 very hard to follow, since some steps are introduced in a different order than how they are performed.
3. It is not clear to me how the model benefits from encoding a field with multiple ones ($\sigma_i$, $\mu_i$, $w_i$, or $z_i$, repeated $G$ times). Intuitively, an encoding step tends to represent a field by *compressing* it into fewer features.
4. Although the linear complexity is mentioned as an advantage, no scaling experiments are provided to support this claim. If no other bottlenecks are present, this architecture can obtain huge advantages in large-scale problems with high spatial resolutions (e.g., industrial-scale 3D problems).

---

> ### Author Rebuttal · Authors · 2026-03-29
>
> We sincerely thank the reviewer for the positive assessment and for recognizing the novelty of the method, the importance of the near-linear complexity, and the value of the interpretability analysis. We understand the main requests as: (i) making the methodology easier to follow, (ii) substantiating the rollout/scaling claims more carefully, and (iii) clarifying several implementation choices. We address each point directly below.
>
> Supplementary tables and figures are at: **https://anonymous.4open.science/r/GPO_rebuttal-BBB1**
>
> **1. Figure 5 / rollout growth rate.**
> Fair point. The figure shows lower absolute error, not unambiguously lower normalized growth rate. We added a normalized rollout analysis (error_t / error_1) in the supplementary. We will revise the main text claim to match what the figure actually supports.
>
> **2. Section 2 clarity.**
> In the revision: (i) all variables defined before first use; (ii) Softplus/Softmax axes stated explicitly in Eqs. (5)/(6); (iii) Sec. 3.2.1 reordered to match actual execution order; (iv) a short pipeline summary added: *Gaussian encoding → basis evaluation → PG measurement → modal coupling → scatter/decode*.
>
> **3. Multiple Gaussians per location.**
> The goal is a localized *mixture* basis, not token compression. A single Gaussian captures one local support pattern; G Gaussians represent overlapping modes with different scales, anisotropies, and weights—critical for fields with fronts, filaments, and locally multi-scale structure. Table 3 in the main paper already shows that increasing G helps up to saturation.
>
> **4. Scaling experiments.**
> Runtime and memory vs. spatial sample count are in the supplementary. The dominant cost grows with G and the N↔G transfers, not quadratically in N, consistent with the near-linear complexity claim.
>
> **Q1. Trivial Gaussian collapse.**
> L_μ anchors weighted centers; L_σ prevents pathological scales; the reconstruction/prediction objective requires nontrivial spatial support to match target fields. Collapse was not observed in practice.
>
> **Q2. 2D/3D joint training.**
> Feasible in principle. The PG operator acts identically on basis coefficients once coordinates are in a consistent format—a 2D input can be embedded into 3D coordinates by padding the third dimension with a constant. The main challenge is balancing sampling density and scale statistics across heterogeneous 2D/3D datasets.
>
> **Q3. Selecting G.**
> Start with moderate G; increase until validation performance saturates. G does not need to scale linearly with resolution since the operator acts in modal space, not physical space. Problems with richer local structure or higher frequency content benefit from larger G.
>
> **Q4. Instabilities with large G.**
> Yes—excessively large G causes optimization instability with diminishing accuracy returns. Performance improves first, then saturates, with increased sensitivity beyond a moderate budget. This motivated our choice.
>
> **Q5. Training strategy.**
> Gaussian Field is pretrained using reconstruction + L2 objectives to establish a stable basis. The operator is then trained jointly with the Gaussian Field (not frozen)—freezing causes structural overfitting to the pretraining objective; joint fine-tuning better adapts the basis to the downstream operator task. We will document explicitly which losses activate in each stage and which modules are updated.
>
> We thank the reviewer again for the constructive suggestions. We believe the added rollout/scaling analyses and the clarifications above substantially improve the paper’s clarity and reproducibility.

---

> > ### Author Rebuttal · Reviewer_stCP · 2026-04-01
> >
> > I thank the authors for their comprehensive answers.

---

### Official Review · Reviewer_YyUg · 2026-03-10

**Soundness:** 3
**Presentation:** 3
**Significance:** 3
**Originality:** 2
**Overall Recommendation:** 4
**Confidence:** 3

**Summary:**

This paper proposes the Gaussian Particle Operator (GPO) for learning dynamics of fluids. Instead of using fixed grid, the authors represent physical fields using a Gaussian basis. They introduce PG Gaussian Attention mechanism to perform latent space encoding and global interactions. The authors reported that the model is extensively evaluated across various benchmarks, demonstrating competitive accuracy while providing intrinsic interpretability.

**Compliance With Llm Reviewing Policy:**

Affirmed.

**Final Justification:**

My final recommendation is *Weak Accept*. I thank the authors for their clear rebuttal. I highly value the paper's practical merits and the extensive empirical evaluation. My initial weaknesses were primarily related to the paper's terminology and theoretical contextualization, and the authors successfully addressed these issues. While the rebuttal resolved these contextual gaps and reinforced my positive assessment, the core technical contribution remains consistent with my initial impression. Therefore, my overall evaluation has not drastically shifted to a higher score, but I consider this a solid, practically useful paper. I support its acceptance.

**Key Questions For Authors:**

## Key Questions For Authors

1. Would you consider discussing your framework within data-driven Koopman operator theory? Your pipeline conceptually is similar to Koopman framework, and acknowledging this connection would enrich the theoretical context.
2. The term particle perhaps might be slightly misleading for readers. Adding a brief discussion to contrast your Eulerian Gaussian basis approach with classical Lagrangian particle methods (like SPH) would clarify the nature of your contribution and prevent misunderstandings.
3. The structural implementation appears reminiscent of the Perceiver architecture. Explicitly discussing this connection in the related work would help readers intuitively understand your computational design. This would be related to the computational cost of the proposed model.
4. Could you either provide insights into how these attention weights might be physically interpreted, or slightly refine your claims to be more precise?

**Limitations:**

yes

**Strengths And Weaknesses:**

Strengths:

- The integration of a local Gaussian basis with a latent attention mechanism is practical and demonstrates engineering merits.
- The empirical evaluation is extensive and impressive. It covers a wide range of benchmarks.
- The ability to directly visualize the learned state representation is a valuable feature for understanding representation.

Weaknesses:

- The overall pipeline reminds me of the framework of data-driven Koopman operators, although the Koopman framework assumes a linear evolution matrix. Also, the term "particle" evokes Lagrangian methods like smoothed particle hydrodynamics (SPH), whereas the proposed method is inherently Eulerian. Discussing the method within this established literature would make the paper more accessible to a broader audience.
- While the Petrov-Galerkin formulation is mathematically sound, the actual architectural implementation might be practically similar to the latent bottleneck structure of the Perceiver architectures. Explicitly mentioning this connection would significantly improve clarity for readers from the deep learning community.
- The text strongly claims an interpretable PDE operator. While the state representation (the Gaussian basis) is indeed interpretable and visualizable, the operator itself (the temporal evolution governed by attention) remains a black box.

Typos:

- In line 215, there is a broken reference: "Eq.(??)". Please correct this.

---

> ### Author Rebuttal · Authors · 2026-03-29
>
> We thank the reviewer for the constructive feedback and we agree with the reviewer that the main issue here is **positioning and claim precision**, not the technical validity of the method. This is especially helpful feedback because it can be addressed directly in the revision. Below we answer the four specific concerns.
>
> Supplementary tables and figures are at: **https://anonymous.4open.science/r/GPO_rebuttal-BBB1**
>
> **1. Relation to data-driven Koopman methods**
>
> We agree there is a meaningful high-level connection: both frameworks use a **lift → evolve → decode** viewpoint. We will add this explicitly in the paper.
>
> At the same time, the difference is important. Koopman-style methods usually aim to find observables whose dynamics are **approximately linear** in latent space. Our goal is different. GPO does **not** seek a linear evolution matrix. Instead, it builds a **nonlinear neural operator** on top of a learned Gaussian basis, with Petrov–Galerkin-style measurement and learned modal coupling. The latent variables are also spatially structured: the basis carries explicit geometry through centers, scales, and weights, and the operator is applied in a resolution-agnostic way on irregular domains as well. So the similarity is conceptual, but the mechanism and objective are not the same. We will add this comparison in related work to make the paper easier to place for readers familiar with Koopman operator learning.
>
> **2. Why we use the term “particle,” and how this differs from SPH**
>
> This is a very fair point. We agree that the word *particle* can easily suggest Lagrangian particle methods such as SPH, while our method is fundamentally **Eulerian**.
>
> In our model, the Gaussian particles are **basis atoms for field representation**. They are not material points that move with the flow, and the method does not simulate particle trajectories or particle interactions in the SPH sense. Concretely, the encoder predicts Gaussian parameters from the field, and the operator mainly updates the **basis coefficients / activations**, not Lagrangian particle states. The field is still queried and reconstructed in an Eulerian manner. We will revise the wording to make this distinction explicit, and we will add one or two sentences contrasting our method with classical Lagrangian particle solvers so that readers from computational physics do not misread the contribution.
>
> **3. Connection to Perceiver-style latent bottlenecks**
>
> At the architectural level, there is indeed a recognizable similarity: both reduce full spatial interaction by routing information through a smaller latent space, which is also the source of the computational savings.
>
> The key difference is that our latent bottleneck is **not a generic learned token set**. The $N \rightarrow G$ step is a **structured Petrov–Galerkin measurement** using Gaussian modal windows, and the same structured windows are used again in the $G \rightarrow N$ scatter step. The latent modes therefore have an explicit numerical interpretation as test / modal functions tied to the Gaussian basis, rather than arbitrary latent tokens. So we agree that Perceiver is the right deep-learning analogy for intuition and complexity, but our construction is more structured than a generic latent bottleneck. We will add this discussion explicitly in related work.
>
> **4. What exactly is interpretable in GPO**
>
> The strongest and most defensible claim is **representation-level interpretability**. The Gaussian basis is directly visualizable through $(\mu, \Sigma, w)$, and these quantities have a clear geometric meaning. For the operator, we only claim a more limited form of interpretability: because modal activations and attention are defined over structured Gaussian modes, we can inspect how energy or activity is redistributed across these modes and map this back to spatial structures. This is useful for diagnosis, but it is not the same as saying the temporal evolution law itself is fully transparent or mechanistically understood. We will revise the wording throughout the paper to reflect exactly this distinction.
>
> **5. Revision details**
>
> We will also fix the broken reference at Eq.(??), and we will update the limitations section accordingly.
>
> Overall, we believe the reviewer’s concerns point to clarification rather than a flaw in the method itself. In the revision we will explicitly position GPO relative to Koopman-style operator learning, distinguish our Eulerian Gaussian basis from SPH-style particle methods, state the Perceiver connection for architectural intuition, and sharpen the interpretability claim so it matches what the method actually provides.

---

> > ### Author Rebuttal · Reviewer_YyUg · 2026-04-01
> >
> > Thank you for your response. My concern has been addressed.

---

### Decision · Program_Chairs · 2026-04-30

**Decision:**

Accept (regular)

**Comment:**

This paper proposes a neural operator that represents fields with a learned Gaussian basis and performs prediction in that basis. Reviewers agreed that the strongest part is the basis-to-basis design itself: it is more original than the median neural operator paper, gives a plausible route to mesh- and resolution-agnostic modeling, and provides intermediate representations that are much more explicit than is usually the case. The main concern is that the empirical case could be stronger. Some baseline comparisons and positioning were questioned, the gains are not always decisive (though I don’t think they always need to be), and the interpretability claim needed to be stated more carefully (here I do agree with the reviewers; I do think this is more interpretable than usual, and a nice original approach, but it needs to be properly qualified.)

The rebuttal clarified several of these points. It explained the ERA5 comparison, added more capacity-fair FNO comparisons and kernel ablations, and narrowed the interpretability claim to representation-level interpretability rather than a one-to-one physical reading. I think the remaining issues are real but mostly about positioning and completeness, not about a fatal weakness in the idea. In a crowded area of neural ops / PDE solvers, this paper is certainly more original and more interesting than average. For these reasons, I recommend acceptance. For the camera-ready, the authors should tighten the claims, sharpen the baseline discussion, and clearly separate reconstruction from prediction throughout.